# Role of Rab13, Protein Kinase A, and Zonula Occludens-1 in Hepatitis E Virus Entry and Cell-to-Cell Spread: Comparative Analysis of Quasi-Enveloped and Non-Enveloped Forms

**DOI:** 10.3390/pathogens13121130

**Published:** 2024-12-20

**Authors:** Shigeo Nagashima, Putu Prathiwi Primadharsini, Masaharu Takahashi, Takashi Nishiyama, Kazumoto Murata, Hiroaki Okamoto

**Affiliations:** Division of Virology, Department of Infection and Immunity, Jichi Medical University School of Medicine, 3311-1 Yakushiji, Shimotsuke-shi 329-0498, Tochigi, Japan; thiwik8@jichi.ac.jp (P.P.P.); mtaka84@jichi.ac.jp (M.T.); nishiyama.takashi@jichi.ac.jp (T.N.); kmurata@jichi.ac.jp (K.M.)

**Keywords:** hepatitis E virus, Rab13, protein kinase A, tight junction protein, zonula occludens-1, HEV entry, HEV cell-to-cell spread

## Abstract

Hepatitis E virus (HEV) exists in two distinct forms: a non-enveloped form (neHEV), which is present in feces and bile, and a quasi-enveloped form (eHEV), found in circulating blood and culture supernatants. This study aimed to elucidate the roles of Ras-associated binding 13 (Rab13) and protein kinase A (PKA) in the entry mechanisms of both eHEV and neHEV, utilizing small interfering RNA (siRNA) and chemical inhibitors. The results demonstrated that the entry of both viral forms is dependent on Rab13 and PKA. Further investigation into the involvement of tight junction (TJ) proteins revealed that the targeted knockdown of zonula occludens-1 (ZO-1) significantly impaired the entry of both eHEV and neHEV. In addition, in ZO-1 knockout (KO) cells inoculated with either viral form, HEV RNA levels in culture supernatants did not increase, even up to 16 days post-inoculation. Notably, the absence of ZO-1 did not affect the adsorption efficiency of eHEV or neHEV, nor did it influence HEV RNA replication. In cell-to-cell spread assays, ZO-1 KO cells inoculated with eHEV showed a lack of expression of HEV ORF2 and ORF3 proteins. In contrast, neHEV-infected ZO-1 KO cells showed markedly reduced ORF2 and ORF3 protein expression within virus-infected foci, compared to non-targeting knockout (NC KO) cells. These findings underscore the crucial role of ZO-1 in facilitating eHEV entry and mediating the cell-to-cell spread of neHEV in infected cells.

## 1. Introduction

The hepatitis E virus (HEV), classified under the genus *Paslahepevirus* within the family *Hepeviridae*, subfamily *Orthohepevirinae* [1], is a small, spherical virus characterized by a single-stranded, positive-sense RNA genome of approximately 7.2 kilobases (kb) in length [2,3]. Its genome features a 5′-untranslated region (UTR) capped with 7-methylguanylate, three open reading frames (ORFs; ORF1, ORF2, and ORF3), and a 3′-UTR with a poly(A) tail [2,3]. ORF1 encodes a non-structural polyprotein comprising several functional domains essential for viral replication, including a methyltransferase (MeT), Y domain, papain-like cysteine protease (PCP), hypervariable region (HVR), X or macro domain, helicase (Hel), and RNA-dependent RNA polymerase (RdRp) [4,5]. ORF2 and ORF3 overlap and are translated from a bicistronic subgenomic RNA measuring 2.2 kb [6]. ORF2 encodes the capsid protein, which exists in three distinct forms: infectious, glycosylated, and cleaved [7,8]. This capsid protein is critical for virion assembly, viral attachment to host cells, and serves as a primary target for neutralizing antibodies [9,10,11]. The glycosylated dimeric form of the ORF2 protein, secreted extracellularly, may function as a decoy to evade the host’s humoral immunity responses during HEV infection [7,8]. The ORF3 protein is a small protein of 112–114 amino acids (aa) located on the HEV virion’s surface and enveloped by a lipid membrane. It plays roles in modulating intracellular signaling pathways, attenuating the host inflammatory response, and providing protection to virus-infected cells [12]. In addition, ORF3 protein is implicated in virion egress from infected cells [13,14,15] and exhibits viroporin activity by forming ion channels [16]. HEV exists in two distinct particle forms: the membrane-unassociated non-enveloped form (neHEV), present in bile and feces, and the membrane-associated quasi-enveloped form (eHEV), which circulates in the bloodstream and is found in culture supernatants [17,18,19,20].

HEV infections are typically self-limiting; however, chronic infection can occur, particularly in immunocompromised individuals [21]. The primary human pathogens are classified under the genus *Paslahepevirus,* specifically the species *Paslahepevirus balayani,* encompassing genotypes 1, 2, 3, and 4 [1]. HEV genotypes 1 (HEV-1) and 2 (HEV-2) are human-specific and predominantly affect populations in developing regions in Asia and Africa, where transmission typically occurs through the consumption of water contaminated with fecal matter [22]. In contrast, HEV genotypes 3 (HEV-3) and 4 (HEV-4) are more prevalent in industrialized nations, with transmission routes that include ingestion of raw or undercooked animal meat products [23,24,25], organ transplantation [21], and blood transfusion [26,27].

Recombinant HEV constructs expressing reporter proteins have been widely utilized in both fundamental and applied virological research [28,29,30,31]. Previously, we engineered an HEV-GLuc replicon, incorporating the *Gaussia* luciferase (GLuc) gene downstream of the ORF2 start codon [32]. In addition, we generated two recombinant infectious HEV variants: one containing the nanoKAZ reporter gene within the ORF1 region (HEV-nanoKAZ) [33], and another featuring a HiBiT tag inserted in the ORF2 region (HEV-HiBiT) [34]. The HEV-nanoKAZ variant facilitates the screening of candidate antiviral agents targeting HEV RNA replication or HEV internalization, while HEV-HiBiT is suitable for screening compounds that inhibit HEV RNA replication, virion assembly, or release [33,34].

The mechanisms underlying each step of the HEV life cycle, particularly the entry processes of both eHEV and neHEV particles, remain poorly characterized. The entry of eHEV and neHEV is initiated by the attachment to host cell membrane proteins, including apolipoprotein E (ApoE), heparan sulfate proteoglycans (HSPGs), asialoglycoprotein receptor (ASGPR), integrin subunit alpha 3 (ITGA3), and adenosine triphosphate synthase subunit 5 beta (ATP5B), facilitating viral internalization [35]. Both eHEV and neHEV particles are subsequently internalized via clathrin-mediated endocytosis [19]. Importantly, eHEV entry further requires the involvement of small guanosine triphosphate (GTPases) Ras-associated binding 5 (Rab5) and 7 (Rab7), which are not required for neHEV entry [19]. On the other hand, previous studies using virus-like particles (VLPs) have demonstrated that internalization occurs through Rab5-positive compartments, relying on dynamin-2, clathrin, and membrane cholesterol [10,36]. Nevertheless, the role of other Rab GTPases in the HEV life cycle remains to be elucidated. Notably, Rab13 has been implicated in the viral entry process of hepatitis C virus (HCV) [37]. Considering the distinct cell entry mechanisms utilized by eHEV and neHEV, a comparative analysis focusing on Rab13 is essential to elucidate the differential pathogenesis of these viral forms.

Protein kinases are critical regulators of immune responses, modulating cellular processes such as protein functions and signal transduction. Viruses exploit these kinase-mediated pathways to facilitate replication. For example, herpes simplex virus replication is contingent on protein kinase A (PKA) activity [38], while the transcription and replication of human immunodeficiency virus (HIV) are enhanced through the combined activation of PKA and protein kinase C (PKC) [39]. In HEV, protein kinase C alpha (PKCα) has been identified as a key host factor that restricts HEV replication [40].

Human viruses employ diverse strategies to ensure effective survival and propagation, including cell-free dissemination, direct cell-to-cell transmission, or a combination of both mechanisms [41]. Direct cell-to-cell transmission typically involves close intercellular contacts that facilitate localized increases in viral particle density, thereby enabling efficient transfer of the virus to adjacent cells [41]. This mode of transmission also provides an advantage by evading antibody-mediated neutralization, thereby enhancing viral dissemination and pathogenesis [42,43,44,45]. Our previous study demonstrated that both eHEV and neHEV particles are abundantly present in the lysates of PLC/PRF/5 cells transfected with wild-type HEV RNA [46]. However, the mechanisms governing HEV cell-to-cell transmission remain poorly characterized, and the cellular factors involved in this process have not yet been identified.

Tight junctions (TJs) are key components of cell-to-cell adhesion complexes, consisting of integral membrane proteins such as occludin (OCLN), claudins (CLDNs), and junctional adhesion molecules (JAMs), which interact with actin cytoskeleton-interacting proteins like zonula occludens (ZO) [47]. Recent studies have highlighted the significance of TJs in the infection processes of various viruses, including roles as viral receptors or co-receptors, as well as in viral internalization, cell-to-cell spread, and egress [48,49,50].

In the present study, we systematically investigated the roles of Rab13, PKA, and TJ proteins in eHEV and neHEV infection using small interfering RNA (siRNA), knockout (KO) cell models, and chemical inhibitors. The findings indicate that HEV infection, regardless of the presence or absence of an envelope on the viral particles, is dependent on Rab13 and PKA for viral entry. In addition, ZO-1 was found to be critical for eHEV entry and for the cell-to-cell spread of neHEV in HEV-infected cells.

## 2. Materials and Methods

### 2.1. Cell Culture

PLC/PRF/5 cells [American Tissue Culture Collection (ATCC, Manassas, VA, USA) no. CRL-8024] were grown in Dulbecco’s modified Eagle medium (DMEM) (Thermo Fisher Scientific, Waltham, MA, USA), supplemented with 10% (vol/vol) heat-inactivated fetal bovine serum (FBS; Thermo Fisher Scientific), referred to as growth medium. Cells were maintained at 37 °C in a humidified incubator with 5% CO_2_. For experiments involving HEV-nanoKAZ-infected cells, HEV-GLuc RNA-transfected cells, HEV-HiBiT RNA-transfected cells, or drug treatments, the growth medium was further supplemented with 1% dimethyl sulfoxide (DMSO) (Fujifilm Wako, Osaka, Japan).

### 2.2. Viruses

HEV particles present in the culture supernatant, derived from a cell culture-adapted HEV-3b JE03-1760F strain (passage 26; 4.3 × 10^7^ copies/mL) [51], were utilized as eHEV for viral inoculation. To generate neHEV, eHEV particles were treated with 0.1% sodium deoxycholate (DOC-Na) and 0.1% trypsin at 37 °C for 3 h, thereby removing the lipid membrane and ORF3 protein from the viral surface, as described previously [20]. In addition, culture supernatants containing neHEV/ΔORF3, a derivative ORF3-deficient mutant (pJE03-1760F/ΔORF3, GenBank accession no. AB437317), in which the ORF3 gene initiation codon was mutated to GCA (Ala) [13], were used for cell-to-cell spread assays.

For luciferase assays, culture supernatants containing eHEV-nanoKAZ (7.1 × 10^7^ copies/mL), generated by transfecting PLC/PRF/5 cells with pJE03-1760F/P10-nanoKAZ3 RNA [33], were utilized. The neHEV-nanoKAZ (4.1 × 10^7^ copies/mL) was prepared by treating eHEV-nanoKAZ particles in the culture supernatant with DOC-Na and trypsin, as described above.

### 2.3. RNA Interference

In this study, the following siRNAs were procured from Dharmacon (Horizon Discovery, Waterbeach, UK): human Rab13 (siGENOME SMARTpool M-008389), human CLDN1 (siGENOME SMARTpool M-017369), human CLDN2 (siGENOME SMARTpool M-020781), human CLDN3 (siGENOME SMARTpool M-017303), human CLDN4 (siGENOME SMARTpool M-013612), human CLDN5 (siGENOME SMARTpool M-011409), human CLDN6 (siGENOME SMARTpool M-015883), human CLDN7 (siGENOME SMARTpool M-004206), human CLDN8 (siGENOME SMARTpool M-017049), human CLDN9 (siGENOME SMARTpool M-014125), human OCLN (siGENOME SMARTpool M-187897), human ZO-1 (siGENOME SMARTpool M-007746), human ZO-2 (siGENOME SMARTpool M-009932), human ZO-3 (siGENOME SMARTpool M-009660), human JAM-A (siGENOME SMARTpool M-005053), human JAM-B (siGENOME SMARTpool M-017389), human JAM-3 (siGENOME SMARTpool M-017350), and control siRNA (siGENOME non-targeting siRNA pool #1 D-001206). PLC/PRF/5 cells were seeded at a density of 1.0 × 10^5^ cells in 24-well plates or 1.5 × 10^4^ cells per well in 96-well plates (Thermo Fisher Scientific) in growth medium. The cells were transfected with 5 nM of siRNA (final concentration) in Opti-MEM (Thermo Fisher Scientific) using DharmaFECT 1 (Horizon Discovery), according to the manufacturer’s instructions.

### 2.4. Western Blotting

After siRNA transfection targeting Rab13, ZO-1, ZO-2, and ZO-3, or a non-targeting control (NC) in PLC/PRF/5 cells, the cells were lysed in lysis buffer [50 mM Tris-HCl (pH 8.0), 1% NP-40, 150 mM NaCl, and a protease inhibitor cocktail (Merck Millipore, Darmstadt, Germany)]. The lysates were subjected to separation via sodium dodecyl sulfate-polyacrylamide gel electrophoresis (SDS-PAGE). The resolved proteins were transferred to polyvinylidene difluoride (PVDF) membranes (0.45 μm; Merck Millipore) and probed with primary antibodies including anti-ORF2 mouse monoclonal antibody (MAb; H6225) [52], anti-ORF3 mouse MAb (TA0536) [17], anti-Rab13 rabbit polyclonal antibody (PAb; Merck Millipore), anti-ZO-1 rabbit PAb (Proteintech, Rosemont, IL, USA), anti-ZO-2 rabbit PAb (Cell Signaling Technology, Danvers, MA, USA), anti-ZO-3 rabbit MAb (Cell Signaling Technology), or anti-β-actin mouse MAb (Fujifilm Wako). Protein bands were visualized using chemiluminescence with the ImageQuant LAS500 system (GE Healthcare, Boston, MA, USA), as previously described [53].

### 2.5. Virus Inoculation of siRNA-Treated Cells

PLC/PRF/5 cell monolayers cultured in 24-well plates and pre-treated with siRNA targeting Rab13, CLDN1 through CLDN9, OCLN, ZO-1, ZO-2, ZO-3, JAM-A, JAM-B, or JAM-C were inoculated with either eHEV or neHEV at 1.0 × 10^5^ copies/well. The cells were incubated for 2 h at 37 °C, washed with phosphate-buffered saline without Mg^2+^ and Ca^2+^ [PBS(−)], and subsequently supplemented with 0.5 mL of growth medium per well. The cultures were maintained at 37 °C, with half of the medium (0.25 mL) being replaced every other day with fresh growth medium. Harvested culture supernatants were centrifuged at 1300× *g* for 2 min at room temperature, and the cells were washed with PBS(−) before being collected in TRIzol reagent (Thermo Fisher Scientific) for RNA extraction. Samples were stored at −80 °C until further analysis.

In addition, PLC/PRF/5 cell monolayers in 96-well plates, pre-treated with siRNA targeting Rab13, were inoculated with eHEV-nanoKAZ (3.0 × 10^6^ copies/well) or neHEV-nanoKAZ (1.0 × 10^6^ copies/well). Following a 4 h incubation at 37 °C, the culture medium was replaced with growth medium containing 1% DMSO, and the cells were incubated at 35.5 °C for 4 days. After incubation, the cells were washed twice with PBS(−) and lysed in 20 µL of cell lysis buffer (JNC Corporation, Tokyo, Japan). The resulting lysates were subsequently analyzed using a luciferase assay, as described below.

### 2.6. Quantification of HEV RNA

Total RNA was extracted from culture supernatants using the TRIzol-LS reagent (Thermo Fisher Scientific) or from cells using the TRIzol reagent. HEV RNA levels were quantified via real-time reverse transcription-polymerase chain reaction (RT-PCR) conducted on a LightCycler apparatus (Roche Diagnostics KK, Tokyo, Japan). The reaction utilized the QuantiTect Probe RT-PCR kit (Qiagen, Tokyo, Japan) with primer sets and a probe targeting the ORF2 and ORF3 overlapping region, as previously described [52].

### 2.7. In Vitro Transcription and Transfection of RNA Transcripts into PLC/PRF/5 Cells

Full-length genomic plasmids, including pHEV3b-GLuc, pHEV3b-GLuc-GAA [32], pHEV3b-HiBiT/ΔORF2s [34], pHEV3b [53], and pHEV3b/ΔORF3 [13], were linearized using the restriction enzyme NheI (New England BioLabs, Ipswich, MA, USA). RNA transcripts were synthesized in vitro using T7 RNA polymerase and the AmpliScribe T7-Flash Transcription kit (Lucigen, Madison, WI, USA). Following transcription, the RNA transcripts were capped with the ScriptCap m7G Capping System (CellScript, Madison, WI, USA). The quality and yield of the synthesized RNAs were verified by agarose gel electrophoresis. Subsequently, 0.1 µg of the capped RNA was transfected into PLC/PRF/5 cells, cultured to >90% confluence in a 96-well plate, using the Xfect RNA transfection reagent (TaKaRa Bio, Shiga, Japan) in accordance with the manufacturer’s protocol. After a 4 h incubation at 37 °C, the culture medium was replaced with 0.1 mL of fresh growth medium containing 1% DMSO or the specified drug concentration (final DMSO concentration of 1%). The cells were incubated at 35.5 °C for 4 days, after which the culture supernatant was collected for luciferase activity measurements.

### 2.8. Luciferase Assay

Intracellular luciferase activity from HEV-nanoKAZ was quantified using h-coelenterazine (h-CTZ) (JNC Corporation) in a specialized assay buffer optimized for CTZ-type luciferase (JNC Corporation), following a previously established protocol [33]. Measurements were conducted using a TriStar2 LB942 multimode plate reader (Berthold Technologies, Bad Wildbad, Germany). For culture supernatants collected from cells transfected with pHEV3b-GLuc RNA, luciferase activity was determined using coelenterazine (JNC Corporation), as described earlier [32]. In addition, for culture supernatants obtained from cells transfected with pHEV3b-HiBiT/ΔORF2s RNA, luciferase activity was assessed with the Nano-Glo HiBiT extracellular detection system (Promega, Madison, WI, USA) according to the manufacturer’s instructions. Prior to measurement, the samples were treated with 0.1% digitonin, as detailed in a previous study [34].

### 2.9. Drug Treatments of PLC/PRF/5 Cells During HEV Infection

Monolayers of PLC/PRF/5 cells cultured in a 24-well plate were washed with PBS(−) and pre-treated for 1 h at 37 °C with various concentrations of the following protein kinase inhibitors, according to the concentrations known to inhibit protein kinases activity [54,55,56,57,58], diluted in growth medium devoid of FBS: H-89 (Cayman Chemical Company, Ann Arbor, MI, USA), myrPKI (Merck Millipore), Ro-31-8220 (Merck Millipore), PD98059 (Merck Millipore), U0126 (Merck Millipore), and SB203580 (Merck Millipore). After treatment, the cells were infected with eHEV or neHEV at a concentration of 1.0 × 10^5^ copies/well at 37 °C for 4 h in the presence and absence of the inhibitors. Non-internalized viruses were subsequently removed, and the cells were washed with PBS(−) and then incubated in growth medium at 37 °C. After 4 days, the cells were washed and then harvested in the presence of TRIzol reagent. The samples were stored at −80 °C until further use.

In a separate experiment, PLC/PRF/5 cells in 96-well plates were inoculated with eHEV-nanoKAZ at 3.0 × 10^6^ copies/well or neHEV-nanoKAZ at 1.0 × 10^6^ copies/well in growth medium without FBS, containing the indicated drugs and a final concentration of 1% DMSO, followed by incubation at 37 °C for 4 h. After incubation, growth medium containing the respective drugs was added to each well with a final concentration of 1% DMSO, and the cells were incubated at 35.5 °C for 4 days.

### 2.10. Cell Viability Assay

The viability of PLC/PRF/5 cells was evaluated using the Cell Counting Kit-8 (Dojindo Laboratories, Kumamoto, Japan) in accordance with the manufacturer’s protocol. Briefly, cells were seeded in 96-well plates and incubated at 37 °C for 2 days. Subsequently, the cells were treated with varying concentrations of the indicated drugs, including a final concentration of 1% DMSO, and incubated at 37 °C for an additional 4 days. A water-soluble tetrazolium salt (WST-8) solution was then added to each well, followed by incubation at 37 °C for 2 h. Absorbance was measured at 450 nm using an iMark microplate reader (Bio-Rad Laboratories, Hercules, CA, USA). The resulting absorbance values were normalized to the vehicle control (1% DMSO).

### 2.11. Semi-Quantitative RT-PCR of TJ Protein Genes

Total RNA was extracted from siRNA-transfected cells using the TRIzol reagent and subsequently used as a template for cDNA synthesis. The synthesis of cDNA was performed with oligo(dT) primer and PrimeScript II RTase (Primescript II 1st strand cDNA synthesis kit; TaKaRa Bio), following the manufacturer’s protocol. PCR amplification of the cDNA was carried out using gene-specific primers (listed in Appendix A) and Platinum SuperFi II DNA polymerase (Thermo Fisher Scientific), according to the manufacturer’s protocol. The resulting PCR products were subjected to agarose gel electrophoresis, and the bands were quantified by densitometric analysis using ImageQuant TL software (ver. 8.1) (GE Healthcare).

### 2.12. Knockout of ZO-1 Using the CRISPR-Cas9 System

ZO-1 gene knockout was performed using the CRISPR-Cas9 system obtained from Dharmacon (Horizon Discovery). PLC/PRF/5 cells were seeded at a density of 8.5 × 10^4^ cells per well in 24-well plates containing growth medium and incubated at 37 °C for 24 h. Transfection was conducted using 50 nM crRNA, 50 nM tracrRNA, and 3 µg of Cas9 plasmid (final concentration) in Opti-MEM (Thermo Fisher Scientific) with DharmaFECT DUO transfection reagent (Horizon Discovery), following the manufacturer’s guidelines. This study utilized the following crRNAs: human ZO-1 crRNA (Edit-R Human ZO-1 crRNA CM-007746-02) and non-targeting control crRNA (Edit-R crRNA Non-targeting Control #1 U-007501). After 48 h of incubation at 37 °C, transfected cells were exposed to 1 µg/mL of puromycin for 72 h to select resistant clones. Indel formation efficiency was analyzed using the Rapid Indel Detection Kit (Nippon Gene, Tokyo, Japan). ZO-1 protein expression in single-cell-derived colonies was confirmed by Western blot analysis. Gene editing was further validated by Sanger sequencing, employing the Guide-it Indel Identification Kit (TaKaRa Bio).

### 2.13. Immunofluorescence Assay

Non-targeting control (NC) or ZO-1 KO PLC/PRF/5 cells, cultured on 2- or 8-well chamber slides (Watson, Tokyo, Japan), were subjected to immunofluorescence analysis. The following primary antibodies were used: anti-claudin-1 rabbit PAb (Merck Millipore), anti-occludin rabbit PAb (Proteintech), anti-ZO-1 rabbit PAb (Proteintech), anti-ZO-2 rabbit PAb (Cell Signaling Technology), anti-ZO-3 rabbit MAb (Cell Signaling Technology), anti-ORF2 mouse MAb (H6225), and anti-ORF3 mouse MAb (TA0536). Secondary antibody incubation was performed using Alexa Fluor 488-conjugated anti-rabbit IgG (Thermo Fisher Scientific) or anti-mouse IgG (Thermo Fisher Scientific) as previously described [53]. Nuclei were stained with 4′,6-diamidino-2-phenylindole (DAPI; Thermo Fisher Scientific). Slides were mounted with Fluoromount/Plus (Diagnostic BioSystems, Pleasanton, CA, USA) and visualized using an FV1000 confocal laser microscope (Olympus, Tokyo, Japan).

### 2.14. Virus Adsorption Assay

Monolayers of PLC/PRF/5 cells cultured in 24-well plates were inoculated with HEV progeny (2.0 × 10^4^ copies/well) and incubated at 4 °C for 1 h to facilitate viral adsorption. After incubation, cells were washed with PBS(−) five times and were lysed using TRIzol reagent. Lysates were stored at −80 °C until further analysis. Neutralization of neHEV particles (2.0 × 10^4^ copies) was performed using anti-ORF2 MAb (H6225; 0.2 mg/mL) or a negative control MAb (MAb 905; 0.2 mg/mL) as previously described [20].

### 2.15. Time-Course Measurement of GLuc Activity

Monolayers of NC or ZO-1 KO PLC/PRF/5 cells cultured in 24-well plates were transfected with 0.5 µg of either pHEV3b-GLuc RNA or pHEV3b-GLuc-GAA RNA. Following a 24 h incubation at 37 °C, the culture medium was replaced with 0.5 mL of fresh growth medium supplemented with 1% DMSO, followed by incubation at 35.5 °C. Supernatants were collected 2, 4, 6, and 8 days post-transfection (dpt) and stored at −80 °C until use. GLuc activity was quantified as described above.

### 2.16. HEV Cell-to-Cell Spread Assay

NC or ZO-1 KO PLC/PRF/5 cells, seeded into a 2-well chamber slide (Watson), were inoculated with eHEV, neHEV, or neHEV/ΔORF3 at a concentration of 5.0 × 10^5^ copies/well, or transfected with pHEV3b RNA or pHEV3b/ΔORF3 RNA at a concentration of 10 ng/well using the TransIT-mRNA transfection kit (TaKaRa Bio), according to the manufacturer’s recommendations. Following a 1 h (virus inoculation) or 4 h (RNA transfection) incubation at room temperature or 37 °C, respectively, the inocula or RNA were removed, and the cells were washed with PBS(−). The culture medium was then replaced with DMEM supplemented with 10% FBS and 1.2% agar (Merck Millipore). After a 4-day (virus inoculation) or 6-day (RNA transfection) incubation at 35.5 °C, the agar overlay was removed, and immunofluorescence staining was performed as described above, using an MAb specific to the viral ORF2 protein (H6225) or ORF3 protein (TA0536).

### 2.17. HEV-nanoKAZ Inoculation in ZO-1 KO Cells

Monolayers of NC or ZO-1 KO PLC/PRF/5 cells cultured in 24-well plates were inoculated with eHEV-nanoKAZ at a concentration of 1.0 × 10^6^ copies/well, or neHEV-nanoKAZ at 5.0 × 10^5^ copies/well [33]. The cells were incubated at 37 °C for 4 h, after which the growth medium with a final concentration of 1% DMSO was added. The cultured cells were maintained at 35.5 °C for 4 days. Following incubation, the cells were washed twice with PBS(−) and lysed with 20 µL of cell lysis buffer (JNC Corporation). Intracellular luciferase activity was subsequently measured.

### 2.18. Statistical Analysis

Data are presented as the mean ± standard deviation (SD). Statistical comparisons were conducted using Student’s *t*-test, with a significance threshold of *p* < 0.05.

## 3. Results

### 3.1. Rab13 Is Important for HEV Entry

To investigate the role of Rab13 in HEV infection, we employed siRNA to deplete Rab13 and assessed its impact on HEV infection within PLC/PRF/5 cells. Cells were treated with 5 nM Rab13-specific siRNA (siRab13) or non-targeting control siRNA (siNC) 3 days prior to and 4 days post virus inoculation (Figure 1A). Following the initial siRNA transfection, the treated cells were inoculated with two distinct forms of HEV particles (eHEV or neHEV) 3 days later, with an inoculum of 1.0 × 10^5^ copies. In eHEV infection, HEV RNA levels in culture supernatants of cells transfected with siNC or no siRNA steadily increased from 6 days post-inoculation (6 dpi), reaching 3.5 × 10^5^ and 3.4 × 10^5^ copies/mL on day 10, respectively (Figure 1B). Conversely, in cells transfected with siRab13, HEV RNA levels in culture supernatants showed minimal increase by day 10, reaching only 9.7 × 10^4^ copies/mL (Figure 1B) (*p* < 0.01). Virus particles generated from Rab13-depleted cells notably decreased to 27.2% compared to those from siNC-transfected cells. In neHEV infection, HEV RNA levels in culture supernatants of cells transfected with siNC or no siRNA reached 1.5 × 10^5^ copies/mL on day 10 (Figure 1C). In contrast, HEV RNA levels in culture supernatants of siRab13-transfected cells were 7.2 × 10^4^ copies/mL, representing 46.7% of those generated from siNC-transfected cells on day 10 (Figure 1C) (*p* = 0.066). In addition, the expression of ORF2 and ORF3 proteins in no siRNA- or siNC-transfected cells inoculated with eHEV or neHEV was detectable at day 10 (Figure 1D,E). In contrast, the expression of ORF2 and ORF3 proteins in siRab13-transfected cells at day 10 was barely detectable (Figure 1D,E). The transfection of siRab13 resulted in a substantial reduction of endogenous Rab13 in the transfected cells, as depicted in Figure 1D,E, whereas no noticeable change was observed in β-actin expression level (days 0 and 10). These findings underscore the Rab13-dependent infectivity of both eHEV and neHEV into PLC/PRF/5 cells.

In this study, three distinct HEV reporter systems (HEV-GLuc, HEV-nanoKAZ, and HEV-HiBiT), recently developed by our team [32,33,34], were employed to ascertain the involvement of Rab13 at specific stages within the HEV life cycle. Initially, we assessed the impact of siRab13 transfection on HEV entry by treating PLC/PRF/5 cells with either 5 nM siRab13 or siNC 3 days prior to eHEV- or neHEV-nanoKAZ inoculation (Figure 2A). After a 4-day incubation period, the infected cells were harvested, and intracellular luciferase activity was quantified. The results depicted in Figure 2B demonstrate a substantial reduction in relative light units (RLUs) within lysates from the Rab13-depleted cells inoculated with eHEV-nanoKAZ (decreased to 33.0%, *p* < 0.001). Similarly, RLU levels notably dropped to 52.0% (*p* < 0.001) in lysates from Rab13-depleted cells inoculated with neHEV-nanoKAZ.

Next, to assess the impact of siRab13 transfection on HEV RNA replication or release, we employed HEV-GLuc or HEV-HiBiT methodologies, respectively. Cells were treated with 5 nM siRab13 or siNC (Figure 2C). Three days after the transfection of each siRNA, the treated cells received pHEV3b-GLuc RNA or pHEV3b-HiBiT/ΔORF2s RNA. Four hours post-transfection at 37 °C, the culture medium was replaced with growth medium and incubated at 35.5 °C and then the culture supernatants were collected after four days (Figure 2C). Luciferase activity in the culture supernatants transfected with pHEV3b-GLuc RNA or pHEV3b-HiBiT/ΔORF2s RNA was assessed directly or after 0.1% digitonin treatment, respectively. The RLU in the culture supernatant of Rab13-depleted cells transfected with pHEV3b-GLuc RNA mirrored that of siNC-transfected cells (Figure 2D, left panel). Similarly, RLUs in the culture supernatant of cells transfected with pHEV3b-HiBiT/ΔORF2s RNA remained unaffected by siRab13 treatment (Figure 2D, right panel). Throughout the observation period, the depletion of endogenous Rab13 persisted, while the expression level of β-actin showed no discernible alterations (Figure 2E). These findings suggest that HEV RNA replication and release were not affected by siRab13 transfection. Taken together, these results distinctly highlight Rab13’s significant role in both the eHEV and neHEV entry steps, exhibiting that eHEV depends more strongly on Rab13 compared to neHEV.

### 3.2. Functional Involvement of Protein Kinase A (PKA) in HEV Entry

According to Köhler [59], there has been documentation of the interaction between Rab13 and PKA, influencing the regulation of TJ assembly. To explore the involvement of protein kinases in HEV infection, we assessed the impact of various established inhibitors: H-89 and myrPKI (PKA inhibitor) [54,55], Ro-31-8220 (PKC inhibitor) [54], PD98059 (MEK1 inhibitor) [56], U0126 (MEK1/2 inhibitor) [57], and SB203580 (p38MEK inhibitor) [58,60]. Preceding inoculation with 1.0 × 10^5^ copies of eHEV or neHEV, PLC/PRF/5 cells were pre-treated with various concentrations of these inhibitors for 1 h and then inoculated in the presence of the inhibitors. After 4 days, the intracellular HEV RNA levels in the cells inoculated with eHEV or neHEV were quantified. The relative levels of HEV RNA in the cells inoculated with eHEV decreased to 64.1%, 49.7%, and 34.9% of that in the untreated control after treatment with 2 µM, 5 µM, and 10 µM H-89, respectively (Figure 3A, left panel) (*p* < 0.01 or *p* < 0.001). Similar reductions were observed in neHEV-inoculated cells: 82.5%, 63.3%, and 56.1% with the respective concentrations of H-89 (Figure 3A, right panel) (*p* < 0.01 or *p* < 0.001). In addition, the intracellular HEV RNA levels in the cells inoculated with eHEV decreased to 67.0%, 37.5%, and 30.4% of that in the untreated control after treatment with 1 µM, 10 µM, and 30 µM myrPKI, respectively (Figure 3B, left panel) (*p* < 0.001). Similar reductions were observed in neHEV-infected cells: 78.4%, 67.6%, and 44.3% with the respective myrPKI concentrations (Figure 3B, right panel) (*p* < 0.05, *p* < 0.01, or *p* < 0.001). In contrast, Ro-31-8220, PD98059, U0126, or SB203580 treatments did not inhibit HEV infectivity compared to untreated controls for either eHEV or neHEV (Figure 3C–F). As shown in Figure 3G,H, H-89 and myrPKI, respectively, had no significant effect on cell viability within 4 days of the drug application, as confirmed by cell viability assay. These findings strongly indicate the necessity of PKA for the infectivity of both eHEV and neHEV.

Next, we employed three distinct HEV reporter systems (HEV-GLuc, HEV-nanoKAZ, and HEV-HiBiT) to ascertain the involvement of PKA at various stages within the HEV life cycle. First, to explore the impact of H-89 or myrPKI treatment on HEV entry, PLC/PRF/5 cells were inoculated with eHEV-nanoKAZ or neHEV-nanoKAZ in the presence of varying concentrations of H-89 or myrPKI (Figure 4A). Intracellular luciferase activity was assessed at 4 dpi. Following treatment with 2 µM, 5 µM, and 10 µM H-89, the RLU in cells inoculated with eHEV-nanoKAZ decreased to 48.3%, 17.9%, and 10.6%, respectively (Figure 4B, left panel) (*p* < 0.001) compared to the untreated control cells. Similarly, RLU in cells inoculated with neHEV-nanoKAZ decreased to 48.2%, 31.4%, and 16.2%, respectively, after treatment with 2 µM, 5 µM, and 10 µM H-89 (Figure 4B, right panel) (*p* < 0.01 or *p* < 0.001). In addition, RLU in cells inoculated with eHEV-nanoKAZ decreased to 69.4%, 56.7%, and 39.9% after treatment with 1 µM, 10 µM, and 30 µM myrPKI, respectively (Figure 4C, left panel) (*p* < 0.05 or *p* < 0.001) compared to the untreated control cells. Similarly, RLU in cells inoculated with neHEV-nanoKAZ decreased to 76.1%, 58.3%, and 44.8% of the levels in the untreated control after treatment with 1 µM, 10 µM, and 30 µM myrPKI, respectively (Figure 4C, right panel) (*p* < 0.05 or *p* < 0.001).

Subsequently, to investigate the effect of H-89 or myrPKI treatment on HEV RNA replication or release, we utilized HEV-GLuc or HEV-HiBiT, respectively. PLC/PRF/5 cells were transfected with pHEV3b-GLuc RNA or pHEV3b-HiBiT/ΔORF2s RNA in the presence of varying concentrations of H-89 or myrPKI (Figure 4D). Four hours after transfection at 37 °C, the culture medium was replaced with growth medium containing various concentrations of H-89 or myrPKI and incubated at 35.5 °C. The culture supernatants were then collected after 4 days (Figure 4D). Luciferase activity in the culture supernatants of the cells transfected with pHEV3b-GLuc RNA or pHEV3b-HiBiT/ΔORF2s RNA was determined directly or after treatment with 0.1% digitonin, respectively. The RLU in the culture supernatants transfected with pHEV3b-GLuc RNA or pHEV3b-HiBiT/ΔORF2s RNA was not affected by treatment with H-89 (Figure 4E, left or right panel, respectively). Similarly, the RLU in the culture supernatants of the cells transfected with pHEV3b-GLuc RNA or pHEV3b-HiBiT/ΔORF2s RNA was not affected by treatment with myrPKI (Figure 4F, left or right panel, respectively). These findings suggest that H-89 and myrPKI treatment did not influence HEV RNA replication or release. Taken together, these results indicate that PKA exerts a crucial role in the entry step of both eHEV and neHEV particles, displaying that eHEV depends more strongly on PKA compared to neHEV.

### 3.3. The TJ Protein Zonula Occludens-1 (ZO-1) Is Important for Growth of Both eHEV and neHEV

In the study by Kohler et al. [59], it was established that Rab13 regulates PKA signaling during TJ assembly. Building upon this, our investigation unveiled the essential roles of Rab13 and PKA in facilitating the entry of both eHEV and neHEV, thereby indicating the potential significance of TJ proteins in HEV infection. Here, to screen the involvement of TJ proteins in HEV proliferation, we employed siRNA targeting CLDN1, 2, 3, 4, 5, 6, 7, 8, 9, OCLN, ZO-1, -2, -3, or JAM-A, -B, -C. Three days after the transfection of specific siRNA or siNC, the treated cells were inoculated with 1.0 × 10^5^ copies of eHEV or neHEV. The cells were collected 4 dpi, and the intracellular HEV RNA levels were quantified. The intracellular HEV RNA levels of the siZO-1-transfected cells inoculated with eHEV reduced to 45.0%, compared to those in the siNC-transfected cells (*p* < 0.01), whereas the downregulation of other TJ proteins showed no discernible effect (Figure 5A). On the other hand, the intracellular HEV RNA levels in the siRNA-transfected cells inoculated with neHEV were similar to those in the siNC-transfected cells, although those in the siZO-1-transfected cells inoculated with neHEV, the HEV RNA levels were reduced to 73.7%, compared to those in the siNC-transfected cells (*p* < 0.001) (Figure 5A). Semi-quantitative RT-PCR analysis showed that siRNA transfection against TJ proteins resulted in a reduction of mRNA levels (16.4−32.2%), as depicted in Figure 5B,C.

To further validate the siRNA screening results, we utilized siRNA to deplete ZO-1, ZO-2, or ZO-3 and examined their effect on virus growth in cultured cells. PLC/PRF/5 cells were treated with 5 nM siZO-1, siZO-2, siZO-3, or NC siRNA 3 days before and 4 days after virus inoculation. Three days after the first transfection of siRNA, the treated cells were inoculated with 1.0 × 10^5^ copies of eHEV or neHEV. The HEV growth kinetic was monitored for 10 days. In both eHEV and neHEV infections, the HEV RNA levels in the culture supernatants of cells transfected with siNC increased gradually from 6 dpi and reached 1.3 × 10^5^ and 1.7 × 10^5^ copies/mL on day 10, respectively (Figure 6A,B). The RNA levels in the culture supernatants of the siZO-2- or siZO-3-transfected cells inoculated with eHEV or neHEV were similar to those of siNC-transfected cells on day 10 (Figure 6A,B, respectively). In sharp contrast, the RNA levels in the culture supernatants of the siZO-1-transfected cells inoculated with eHEV or neHEV significantly decreased to 11.4% and 13.3% of those released from cells transfected with siNC, respectively (Figure 6A,B) (*p* < 0.01). The expression of ORF2 and ORF3 proteins in the siNC-, siZO-2-, or siZO-3-transfected cells inoculated with eHEV or neHEV was detectable at day 10 (Figure 6C,D). In contrast, the expression of ORF2 and ORF3 proteins in the siZO-1-transfected cells at day 10 was undetectable (Figure 6C,D). In addition, transfection of siZO-1, siZO-2, or siZO-3, but not siNC, caused a marked reduction in the respective levels of endogenous ZO-1, ZO-2, or ZO-3 protein in the transfected cells inoculated with eHEV or neHEV (days 0 and 10), respectively (Figure 6C,D). In contrast, no discernible alterations were observed in the expression levels of β-actin. These findings collectively emphasize the critical role of ZO-1 in the growth of both eHEV and neHEV.

### 3.4. Characterization of ZO-1 Knockout (KO) PLC/PRF/5 Cells

To assess ZO-1’s impact on the infectivity of both eHEV and neHEV, we silenced its expression using the CRISPR-Cas9 system. After single-cell cloning, cells underwent screening via Western blotting and immunofluorescence assay to identify the KO cell. In the left panel of Figure 7A, ZO-1 was absent in the cell lysate of ZO-1 KO cells, while detectable in the wild-type (WT) or NC KO cells via Western blotting. Conversely, β-actin showed consistent levels across WT, NC KO, and ZO-1 KO cells (Figure 7A, right panel). Immunofluorescence staining revealed ZO-1 expression at intercellular junctions in NC KO cells (Figure 7B, left panel), contrasting its absence in ZO-1 KO cells (Figure 7B, right panel). Nucleotide sequencing of the PCR product against the target region confirmed successful biallelic KO of 1 (at 2 distinct positions) or 13 nucleotides. Since ZO-1 plays an important role in forming and maintaining TJs in polarized cells [61], we evaluated the cellular localization of CLDN1, OCLN, ZO-2, and ZO-3 in ZO-1 KO cells, finding no impact on the subcellular localization of these proteins (Figure 7C).

To validate these results using siZO-1, ZO-1 KO cells were inoculated with 1.0 × 10^5^ copies of eHEV or neHEV. Cells were collected 4 dpi, and intracellular HEV RNA levels were quantified. In ZO-1 KO cells inoculated with eHEV or neHEV, intracellular HEV RNA levels reduced to 6.4% or 24.6%, respectively, compared to those in NC KO cells (Figure 7D) (*p* < 0.001). These findings suggest that ZO-1 KO cells can serve as tools for assessing the impact of ZO-1 on HEV infection.

### 3.5. Analysis of HEV Growth Using ZO-1 KO PLC/PRF/5 Cells

To examine the infectivity of eHEV and neHEV on ZO-1 KO cells, both forms of HEV particles were inoculated into ZO-1 KO cells. First, quantification of HEV RNA levels in the culture supernatants was performed using real-time RT-PCR (Figure 8A,B). In NC KO cells inoculated with eHEV or neHEV, the RNA levels in the culture supernatants increased starting from 4 dpi, reaching 1.9 × 10^8^ and 1.8 × 10^8^ copies/mL, respectively, by 16 dpi (Figure 8A,B). In sharp contrast, RNA levels in the culture supernatants of ZO-1 KO cells inoculated with eHEV or neHEV showed no increase even up to 16 dpi (Figure 8A,B).

Further, to compare the adsorption efficiency of eHEV or neHEV to NC KO or ZO-1 KO cells, the quantity of adsorbed virus was measured using real-time RT-PCR 1 h after inoculation. The HEV RNA levels in ZO-1 KO cells inoculated with eHEV were comparable to those in NC KO cells (Figure 8C, left panel). Similarly, HEV RNA levels in ZO-1 KO cells inoculated with neHEV were akin to those in NC KO cells (Figure 8C, right panel). In contrast, the HEV RNA levels decreased in both NC KO cells and ZO-1 KO cells inoculated with neHEV treated with MAb H6225—which targets the HEV capsid protein and neutralizes neHEV—compared to the RNA levels in cells inoculated with neHEV treated with the unrelated negative control (MAb 905) (Figure 8D). These findings suggest that the disruption of ZO-1 did not impact the adsorption of either eHEV or neHEV particles.

To bypass the HEV entry step and exclusively assess HEV RNA replication, we utilized the HEV-GLuc. GLuc activity in the culture supernatants of the pHEV3b-GLuc RNA-transfected NC KO cells increased until 8 dpt (Figure 8E). A similar level of GLuc activity was detected in ZO-1 KO cells transfected with pHEV3b-GLuc RNA until 8 dpt (Figure 8E). Conversely, GLuc activity did not increase in the culture supernatants of NC or ZO-1 KO cells transfected with RNA transcripts of a replication-defective mutant (pHEV3b-GLuc-GAA). These results indicate that ZO-1 did not influence the HEV RNA replication.

### 3.6. Intercellular Spread of HEV via Cell-to-Cell Contact

It has been reported that TJ proteins are involved in the spread of viruses via cell-to-cell contact [48,49,50]. Subsequently, we investigated the involvement of ZO-1 in the intercellular spread of HEV. The spread of HEV to the surrounding cells in PLC/PRF/5 cells was observed using a monolayer cell culture overlaid with culture medium containing 1.2% agarose, a conventional method for virus titration as the plaque assay. Four days after inoculation with eHEV or neHEV, followed by overlaying with agarose, the monolayers of NC KO or ZO-1 KO PLC/PRF/5 cells were subjected to immunofluorescence staining using anti-ORF2 or anti-ORF3 MAbs (Figure 9A,B, respectively). In NC KO cells, the expression of ORF2 and ORF3 proteins was detected in cells inoculated with eHEV or neHEV, confirming the presence of typical HEV-infected foci. Conversely, in ZO-1 KO cells inoculated with eHEV, the expression of HEV ORF2 and ORF3 proteins was not detected (Figure 9A,B). These findings indicate that ZO-1 is necessary for the initial entry of eHEV particles. Meanwhile, in neHEV-infected cells, the observable expression of ORF2 and ORF3 proteins within the virus-infected foci in ZO-1 KO cells was notably lower than that in NC KO cells (Figure 9A,B). The efficiency of neHEV adsorption and RNA replication in ZO-1 KO cells was similar to that in NC KO cells (Figure 8C,E, respectively), suggesting the involvement of ZO-1 in the cell-to-cell spread of HEV.

Previously, we reported the presence of eHEV particles within the multivesicular body (MVB) with internal vesicles and the association of HEV release with the exosomal pathway, while neHEV particles were found in the cytoplasm [46,62]. To ascertain the particle involved in cell-to-cell spread, we utilized an ORF3-deficient variant (neHEV/ΔORF3), which lacked eHEV particle production due to the absence of ORF3 protein [13]. Four days after inoculation with neHEV/ΔORF3 and agarose overlay, the monolayers of NC KO or ZO-1 KO PLC/PRF/5 cells were subjected to immunofluorescence staining with anti-ORF2 MAb. In NC KO cells, typical HEV-infected foci, indicated by detectable ORF2 protein expression, were observed in cells inoculated with neHEV/ΔORF3. In contrast, ZO-1 KO cells inoculated with neHEV/ΔORF3 showed notably fewer detectable ORF2 expressions within the virus-infected foci compared to NC KO cells (Figure 9C). These results suggest the participation of neHEV in the cell-to-cell spread of HEV.

To determine whether the observed differences in the size of virus-infection foci in ZO-1 KO cells inoculated with neHEV were independent of the viral entry step, we conducted a parallel investigation using RNA transfection. RNA transfection bypasses the entry step, enabling the direct evaluation of post-entry events, including viral RNA replication. In the cells transfected with wild-type HEV RNA (pHEV3b RNA), both eHEV and neHEV were produced intracellularly. Notably, the expression of ORF2 protein within the virus-infection foci in ZO-1 KO cells was significantly lower compared to NC KO cells (Figure 10, upper panels). Similarly, in cells transfected with ORF3-deficient variant RNA (pHEV3b/ΔORF3 RNA), which exclusively produces neHEV intracellularly, the expression of ORF2 protein within the virus-infection foci in ZO-1 KO cells was also significantly lower than that in NC KO cells, consistent with the findings from wild-type RNA transfection (Figure 10, lower panels). These results collectively suggest that the observed differences in the size of the virus-infection foci are attributable to viral transmission processes rather than the entry step of HEV infection.

### 3.7. Functional Involvement of ZO-1 in the HEV Entry Step

To elucidate the role of ZO-1 in the internalization of HEV, we utilized eHEV-nanoKAZ and neHEV-nanoKAZ reporter systems. NC KO or ZO-1 KO cells were inoculated with cell culture-derived eHEV-nanoKAZ (3.0 × 10^6^ copies/well) or neHEV-nanoKAZ (1.0 × 10^6^ copies/well). Subsequently, infected cells were collected after 48 h, 72 h, and 96 h, and the intracellular luciferase activity was measured using h-CTZ as a substrate. In the case of neHEV-nanoKAZ infection, luciferase activity in ZO-1 KO cells was comparable to that in NC KO cells at 48 h post-infection. However, at 72 and 96 h, luciferase activity in ZO-1 KO cells decreased to an average of 72.9% and 42.9% of that observed in NC KO cells, respectively (*p* < 0.05 and *p* < 0.001) (Figure 11). These findings indicate that ZO-1 is not critical for the entry of neHEV but plays a significant role in subsequent post-entry events. This conclusion aligns with the results obtained from the cell-to-cell spread assay (Figure 10). In contrast, eHEV-nanoKAZ infection showed a pronounced dependence on ZO-1 for successful viral entry. Luciferase activity in ZO-1 KO cells decreased markedly to an average of 9.3%, 4.6%, and 14.0% of NC KO levels at 48, 72, and 96 h post-infection, respectively (*p* < 0.001 for all time points). Together, these data highlight the distinct functional requirements for ZO-1 in the entry and post-entry processes of HEV, revealing its essential role in eHEV entry while being partially dispensable for neHEV internalization.

## 4. Discussion

The present study demonstrated that Rab13, PKA, and the TJ protein ZO-1 are essential components in the HEV life cycle. Using three distinct HEV reporter systems (HEV-GLuc, HEV-nanoKAZ, and HEV-HiBiT), we found that ZO-1 is particularly critical for the entry of eHEV particles, while it does not affect other stages of the viral life cycle. Furthermore, we showed that ZO-1 facilitates the cell-to-cell spread of neHEV particles.

Viruses often exploit host cellular machinery to complete various stages of their life cycles. The Rab GTPases family is the largest group of small GTPases involved in vesicular transport, with nearly 70 Rab family members identified [63]. Rab GTPases are known to play roles in the replication of several key human viral pathogens, including HIV, HCV, and herpesviruses [64]. Specifically, Rab5, Rab7, Rab11, and Rab27 have been implicated in HEV propagation [19,62,65]. However, it remains uncertain whether other Rab GTPases are involved in HEV proliferation. We first investigated the role of Rab13 in HEV infection by employing siRNA-mediated knockdown. Our results indicated that Rab13 is required for the infectivity of both eHEV and neHEV in PLC/PRF/5 cells (Figure 1B,C). Notably, siRab13 transfection did not affect HEV RNA replication or viral release, but it significantly inhibited the entry of both eHEV and neHEV (Figure 2). This finding is consistent with observation in HCV, where Rab13 has been shown to play an important role in the viral entry step, without being necessary for HCV RNA replication [37]. Although CLDNs and OCLN are essential for HCV entry into host cells [49], and their membrane trafficking is regulated by Rab13 [66,67], our data demonstrated that depletion of CLDNs and OCLN did not impair the entry of eHEV and neHEV (Figure 5A). Thus, further investigation is warranted to elucidate the mechanisms by which Rab13 regulates HEV entry.

Protein kinases are known to play crucial roles in the life cycle of various viruses, including HCV, adenovirus, herpes simplex virus, influenza virus, and HIV [38,39,68,69,70]. In the present study, we screened a panel of established kinase inhibitors to evaluate their impacts on HEV infection and identified PKA as a key factor involved in the entry of both eHEV and neHEV. Treatment of PLC/PRF/5 cells with PKA inhibitors, such as H-89 or myrPKI, significantly inhibited the infection of both eHEV and neHEV (Figure 3). Furthermore, the reduction in luciferase activity was observed exclusively in the HEV-nanoKAZ assay following drug treatment, indicating that active PKA expression in target cells is essential for HEV entry (Figure 4). On the other hand, previous studies have identified PKCα as an important host factor that restricts HEV replication [40]. Both the use of specific PKCα inhibitors and short hairpin RNA (shRNA)-mediated knockdown of PKCα enhanced HEV replication, whereas overexpression of the activated form of PKCα or treatment with its pharmacological activator markedly inhibited viral replication. However, in our study, the PKC inhibitor Ro-31-8220 did not significantly affect HEV infectivity even in the reported inhibition concentrations (Figure 3C) [54]. This discrepancy may be attributed to differences in experimental approaches between the studies, such as the use of wild-type virus versus replicon systems.

TJs are highly specialized membrane domains that play critical roles in various cellular processes [71]. Recent research has highlighted the significant involvement of TJs in the infection processes of several viruses. For example, TJ proteins such as JAM-A or certain members of the claudin family are used as receptors or co-receptors by viruses from the *Sedoreoviridae* family and HCV during host cell entry [49]. In addition, JAM-A, OCLD, and ZO-1 have been shown to be crucial for rotavirus internalization [48]. In the present study, we investigated the role of TJ proteins in the HEV infection. Specific knockdown of ZO-1 significantly impaired the infection of both eHEV and neHEV (Figure 5 and Figure 6). In addition, in ZO-1 KO cells inoculated with either eHEV or neHEV, viral RNA levels in the culture supernatant showed no significant increase up to 16 dpi (Figure 8A,B). We previously developed HEV-nanoKAZ, which facilitates the convenient monitoring of the HEV entry process [33]. In the present study, we demonstrated that ZO-1 specifically mediates the entry step of HEV in the HEV-nanoKAZ assay (Figure 11), following the results from the cell-to-cell spread assay (Figure 9 and Figure 10). In contrast, ZO-1 KO did not affect the adsorption efficiency of either eHEV or neHEV (Figure 8C), nor did it alter HEV RNA replication (Figure 8E). Consequently, other TJ proteins did not appear to be involved in the entry of eHEV or neHEV in PLC/PRF/5 cells, indicating that TJ proteins do not function as receptors or co-receptors during HEV infection.

Viruses can spread through two distinct mechanisms: cell-free particle transmission and direct cell–cell contact, also referred to as cell-to-cell transmission [41]. Previous studies have demonstrated that HIV-1 spreads more rapidly via cell-to-cell transmission compared to cell-free transmission [72,73]. In the context of HEV infection, Oshiro et al. reported that in human hepatocytes, HEV may propagate predominantly through cell-to-cell transmission across the cell membrane rather than via the extracellular culture medium [74]. To investigate the distinct modes of virus transmission, various approaches have been employed in cell culture models. Experimental setups have been designed to selectively inhibit either cell-free or cell-to-cell spread, enabling the determination of their respective contribution to overall infection dynamics. For example, the addition of viscous substances such as methylcellulose or agarose to infected cultures impedes the diffusion of extracellular viral particles, thereby suppressing cell-free transmission while leaving cell-to-cell transmission unaffected [75,76,77,78]. In the present study, we used agarose to analyze cell-to-cell spread and found that ZO-1 expression is essential for the efficient cell-to-cell transmission of neHEV, as evidenced by using ZO-1 KO cells (Figure 9 and Figure 10). Our previous work established that both eHEV and neHEV particles are present within infected cells; in particular, the eHEV particles are present within the MVB and are released via the exosomal pathway, while the neHEV particles are found in the cytoplasm [46,62]. In the present study, we further elucidated that these two forms of HEV utilize distinct pathways for virus dissemination: eHEV is released into the culture medium, facilitating spread via cell-free transmission, while neHEV spreads via direct cell-to-cell contact. This differential utilization of transmission routes may represent a unique strategy employed by HEV for viral dissemination in cell culture.

Transmission via cell-to-cell contacts is considered to contribute to immune evasion by shielding the virus from neutralizing antibodies [79,80] and reduce the efficacy of antiviral drug therapies owing to the high local concentration of the virus [81]. eHEV particles released from hepatocytes are believed to circulate in the bloodstream, playing a key role in viral replication within the host [18]. Furthermore, we demonstrated that the infectivity of eHEV particles was not compromised by the presence of neutralizing antibodies in the culture supernatant [20,34]. In this study, we revealed that neHEV efficiently spreads via direct cell-to-cell transmission (Figure 9 and Figure 10), suggesting that this mechanism may serve as a viral strategy to circumvent neutralizing antibodies in the bloodstream and facilitate sustained infection.

An investigation into the role of ZO-1 in HEV entry revealed no significant increase in intracellular luciferase activity between 48 and 96 h in ZO-1 KO cells inoculated with eHEV-nanoKAZ (Figure 11). In contrast, intracellular luciferase activity in ZO-1 KO cells inoculated with neHEV-nanoKAZ was comparable to that observed in NC KO cells at 48 h (Figure 11). These results suggest that ZO-1 is essential for the internalization of eHEV but dispensable for the internalization of neHEV. Notably, intracellular luciferase activity in neHEV-nanoKAZ-infected cells decreased over time (Figure 11). Further analysis using a cell-to-cell spread assay via RNA transfection revealed that ZO-1 facilitates the cell-to-cell spread of neHEV particles (Figure 10). However, the specific roles of ZO-1 in later stages of the HEV life cycle, including viral assembly and release, remain unclear. Addressing these gaps will be a focus of future studies.

In conclusion, our data provided evidence for the functional involvement of Rab13 and PKA in the entry of both eHEV and neHEV. Furthermore, the present study revealed the critical role of ZO-1 in facilitating eHEV entry and mediating the cell-to-cell spread of neHEV within infected cells. These insights contribute to the understanding of host factors essential for HEV propagation and may inform the development of novel antiviral targets.

## Figures and Tables

**Figure 1 pathogens-13-01130-f001:**
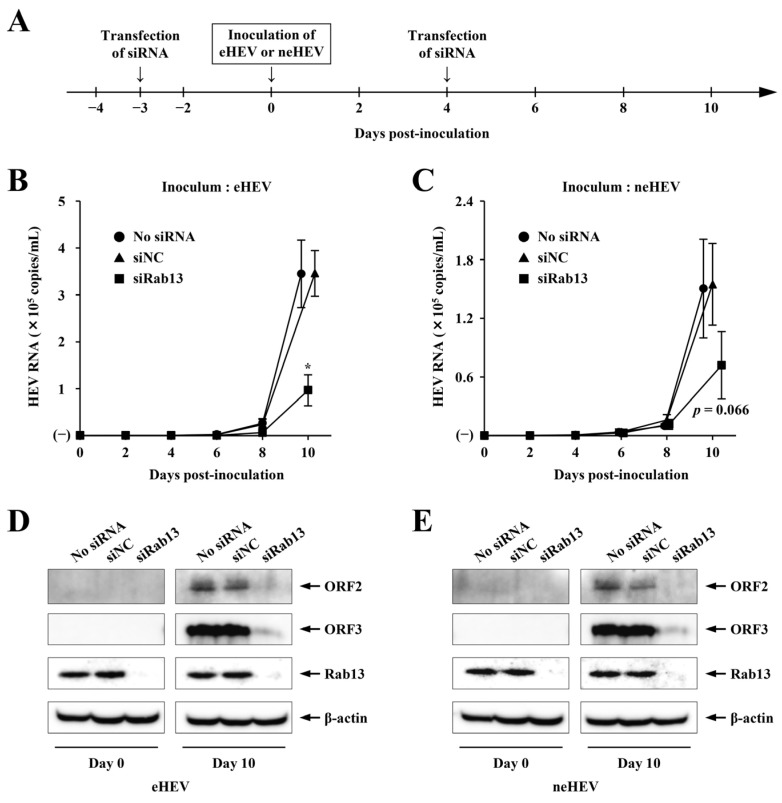
Impact of Rab13 silencing on HEV infection. (**A**) Schematic of the experimental protocol. PLC/PRF/5 cells were transfected with siRNA at two time points: 3 days prior to and 4 days post-HEV inoculation. HEV replication was monitored for up to 10 days post-inoculation (dpi). (**B**,**C**) Quantification of HEV RNA in the culture supernatants of cells inoculated with eHEV (**B**) or neHEV (**C**) and treated with siRab13. Data are presented as the mean ± standard deviation (SD) from three independent experiments. * *p* < 0.01. (**D**,**E**) Intracellular expression of HEV ORF2 and ORF3 proteins and knockdown efficiency of Rab13-specific siRNA (siRab13). PLC/PRF/5 cells were transfected with siRab13, non-targeting control siRNA (siNC), or buffer only (no siRNA). Cells were lysed on the indicated days following inoculation with eHEV (**D**) or neHEV (**E**), and the expression levels of ORF2 protein, ORF3 protein, Rab13, and β-actin were assessed by Western blotting using anti-ORF2 MAb, anti-ORF3 MAb, anti-Rab13 PAb, and anti-β-actin MAb, respectively.

**Figure 2 pathogens-13-01130-f002:**
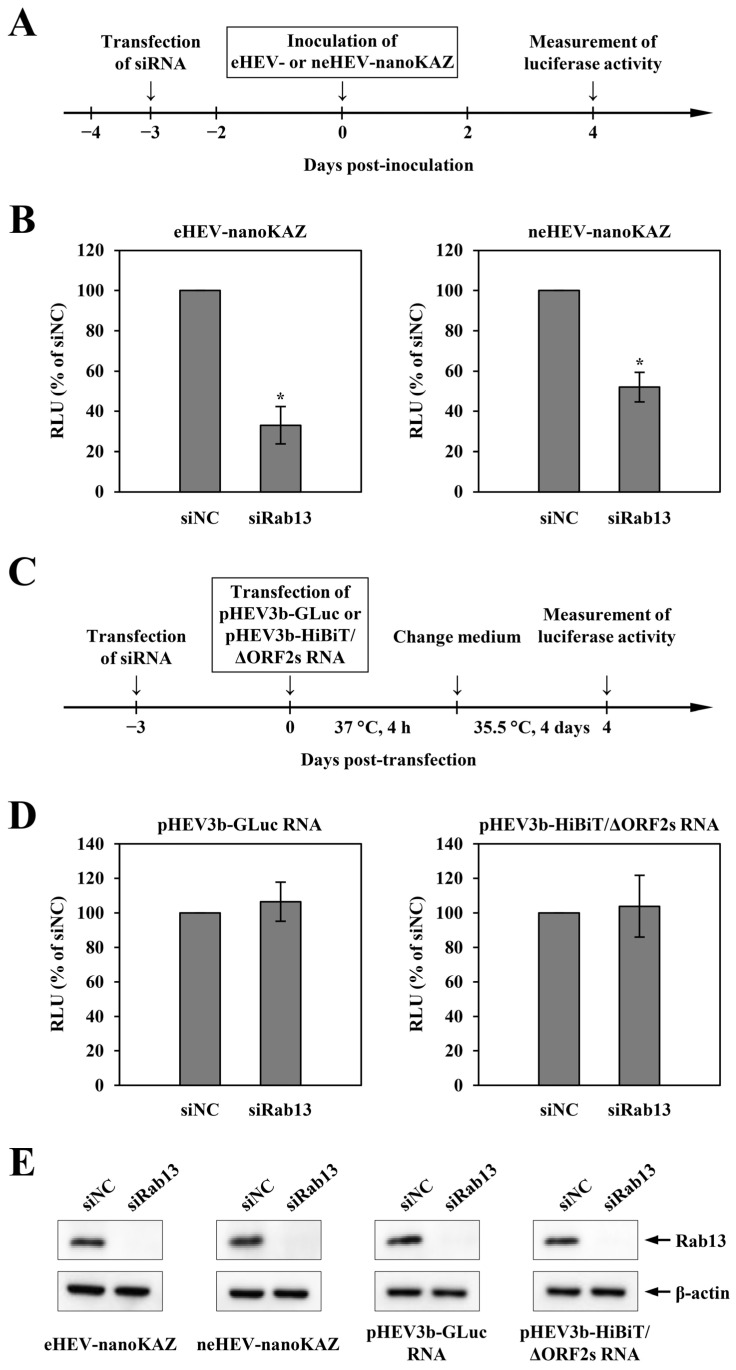
The role of Rab13 in the HEV life cycle. (**A**) Schematic of the experimental design for the HEV-nanoKAZ assay. PLC/PRF/5 cells were transfected with siRNA 3 days prior to virus inoculation. Cells were then infected with eHEV-nanoKAZ or neHEV-nanoKAZ, and cell lysates were collected 4 dpi. (**B**) Intracellular luciferase activity in siRab13-treated cells inoculated with eHEV-nanoKAZ (**left panel**) or neHEV-nanoKAZ (**right panel**) at 4 dpi. Luciferase activity was measured using h-coelenterazine. (**C**) Schematic of the experimental design for the HEV-GLuc and HEV-HiBiT assays. PLC/PRF/5 cells were transfected with siRNA 3 days before transfection with pHEV3b-GLuc RNA or pHEV3b-HiBiT/ΔORF2s RNA. Following a 4 h inoculation at 37 °C, the culture medium was replaced with fresh growth medium, and the cells were incubated at 35.5 °C. Culture supernatants were collected 4 days post-transfection (dpt). (**D**) Luciferase activity in the culture supernatants of siRab13-treated cells transfected with pHEV3b-GLuc RNA (**left panel**) or pHEV3b-HiBiT/ΔORF2s RNA (**right panel**) at 4 dpt. Luciferase activity derived from GLuc was measured using coelenterazine, while HiBiT activity was assessed with the Nano-Glo HiBiT extracellular detection system after treatment of the culture supernatant with 1% digitonin. (**E**) Efficiency of Rab13 knockdown by siRNA. Cells were lysed 4 days after inoculation with eHEV or neHEV, or 4 days after transfection with pHEV3b-GLuc RNA or pHEV3b-HiBiT/ΔORF2s RNA. Expression levels of Rab13 (**upper panels**) and β-actin (**lower panels**) were analyzed by Western blotting using anti-Rab13 PAb and anti-β-actin MAb, respectively. All data are presented as the mean ± SD of three independent experiments. * *p* < 0.001.

**Figure 3 pathogens-13-01130-f003:**
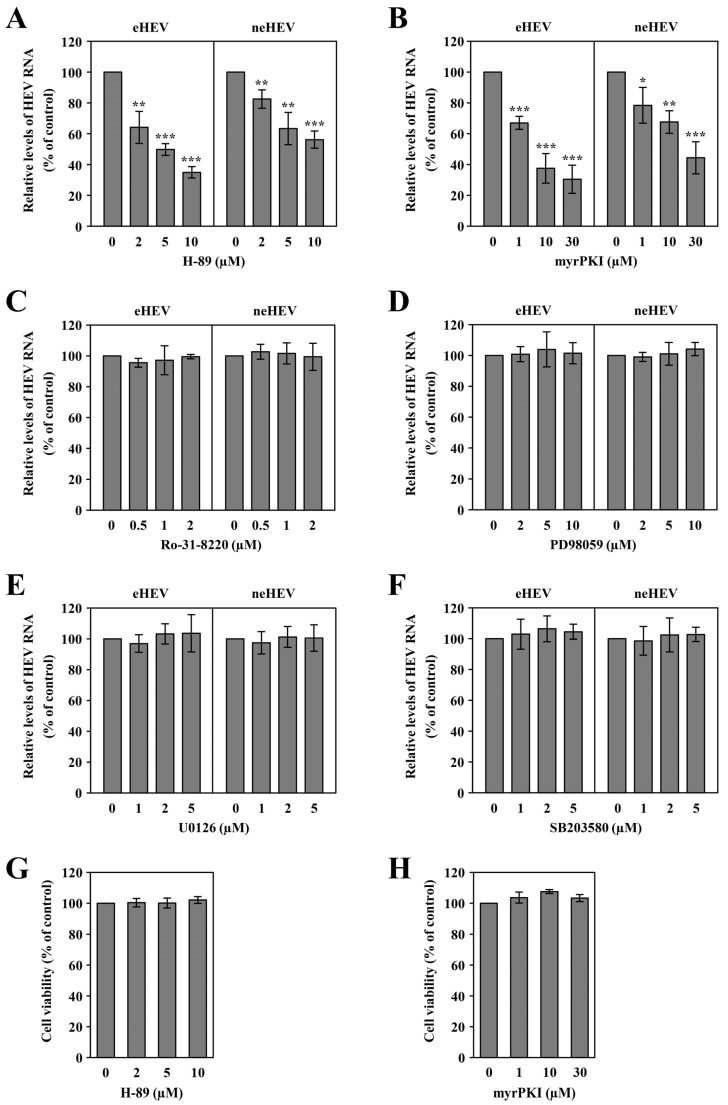
Impact of protein kinase inhibitors on HEV infection. PLC/PRF/5 cells inoculated with eHEV or neHEV were cultivated in growth medium containing the indicated inhibitors, and cells were harvested at 4 dpi. (**A**–**F**) Intracellular HEV RNA levels in eHEV-infected (**left panels**) or neHEV-infected cells (**right panels**) treated with H-89 (**A**), myrPKI (**B**), Ro-31-8220 (**C**), PD98059 (**D**), U0216 (**E**), or SB203580 (**F**) at 4 dpi. Data are presented as the mean ± SD of three independent experiments. * *p* < 0.05, ** *p* < 0.01, *** *p* < 0.001, indicating statistical significance. (**G**,**H**) Cell viability assays conducted to assess cellular proliferation and survival following treatment with various concentrations of H-89 (**G**) or myrPKI (**H**), performed 4 days after drug treatment. Data represent the mean ± SD of two independent experiments.

**Figure 4 pathogens-13-01130-f004:**
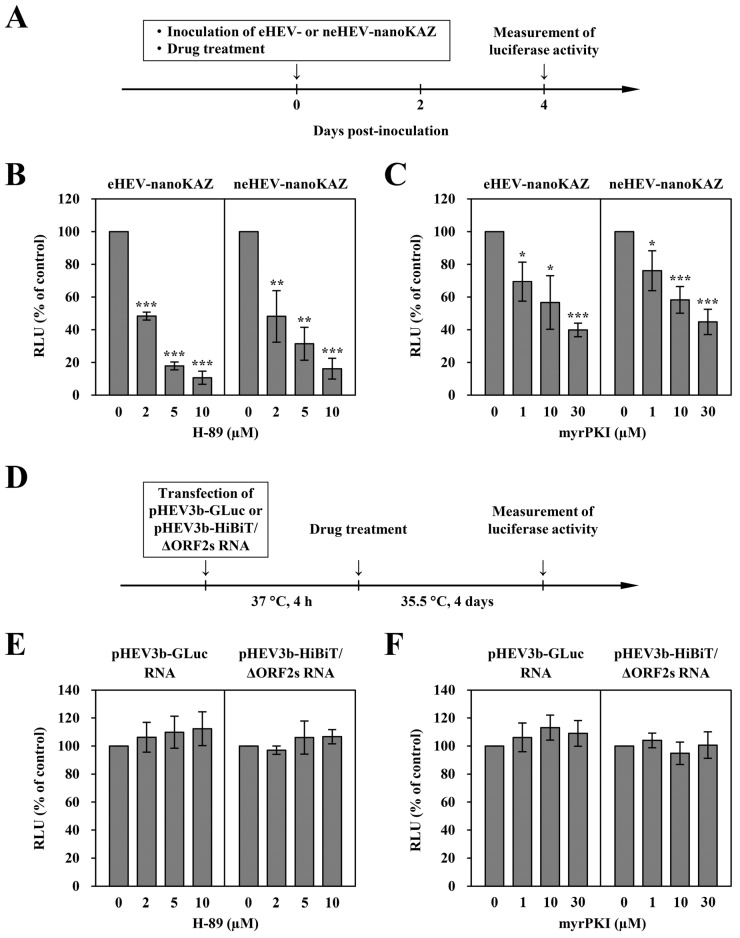
Identification of HEV life cycle stages inhibited by PKA inhibitors, H-89 or myrPKI. (**A**) Schematic of the experimental design for the HEV-nanoKAZ assay. PLC/PRF/5 cells were inoculated with either eHEV-nanoKAZ or neHEV-nanoKAZ in the presence of the indicated inhibitors, and cell lysates were collected 4 dpi. (**B**,**C**) Intracellular luciferase activity measured in cells treated with H-89 (**B**) or myrPKI (**C**), following inoculation with eHEV-nanoKAZ (**left panels**) or neHEV-nanoKAZ (**right panels**) at 4 dpi. (**D**) Experimental timeline for the HEV-GLuc and HEV-HiBiT assays. PLC/PRF/5 cells were transfected with pHEV3b-GLuc RNA or pHEV3b-HiBiT/ΔORF2s RNA. After a 4 h incubation at 37 °C, the culture medium was replaced with fresh growth medium containing the indicated inhibitor, and cells were incubated at 35.5 °C. Culture supernatants were collected 4 dpt. (**E**) Luciferase activity in the culture supernatants of cells transfected with pHEV3b-GLuc RNA (**left panel**) or pHEV3b-HiBiT/ΔORF2s RNA (**right panel**) at 4 dpt, following treatment with H-89. (**F**) Luciferase activity in the culture supernatants of cells transfected with pHEV3b-GLuc RNA (**left panel**) or pHEV3b-HiBiT/ΔORF2s RNA (**right panel**) at 4 dpt, following treatment with myrPKI. Data are presented as mean ± SD from three independent experiments. * *p* < 0.05, ** *p* < 0.01, *** *p* < 0.001.

**Figure 5 pathogens-13-01130-f005:**
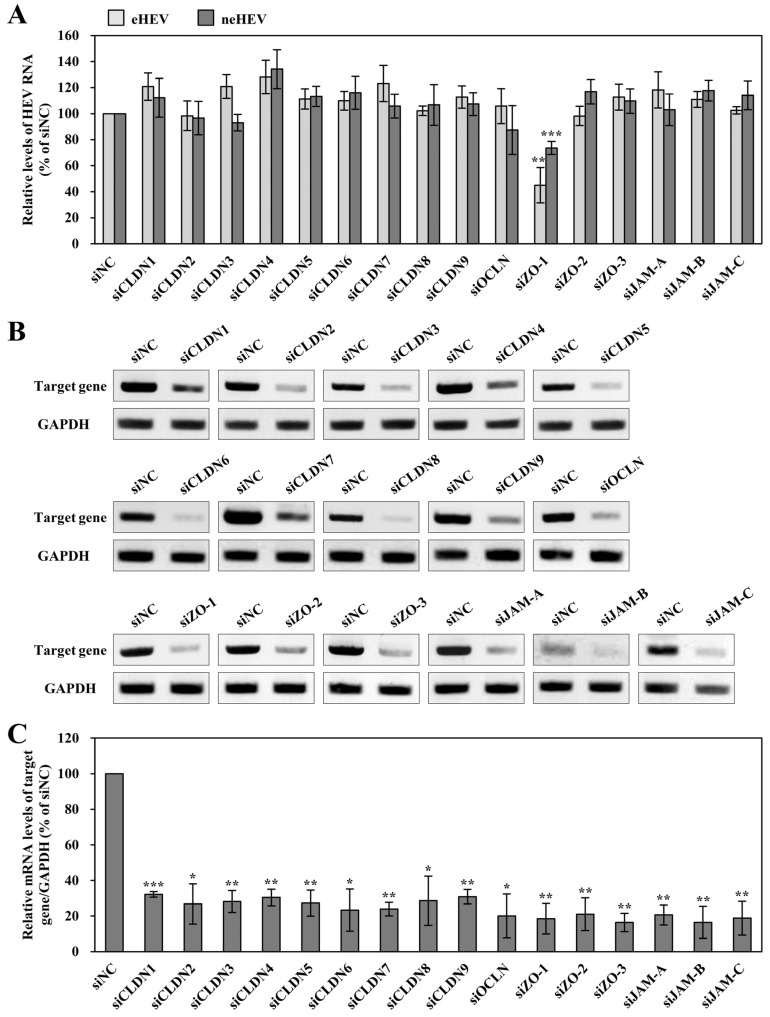
Role of tight junction (TJ) proteins in HEV infection. (**A**) Intracellular HEV RNA titer in eHEV- or neHEV-infected PLC/PRF/5 cells treated with siRNA. Cells were transfected with siRNA 3 days prior to virus inoculation. Subsequently, eHEV or neHEV was inoculated to the siRNA-treated cells, and cell lysates were collected 4 dpi. Data are presented as mean ± SD from three independent experiments. (**B**,**C**) The mRNA expression levels of siRNA specific for TJ-protein transfected cells analyzed by semi-quantitative RT-PCR. Cell lysates were collected 4 days after inoculation. Each representative gel shows the target gene (**upper panels**) and GAPDH as the control (**lower panels**) (**B**). The mRNA expression of TJ protein is presented as a percentage standardized against target gene mRNA expression levels in siNC-transfected cells (**C**). Data are presented as mean ± SD from two independent experiments. * *p* < 0.05, ** *p* < 0.01, *** *p* < 0.001.

**Figure 6 pathogens-13-01130-f006:**
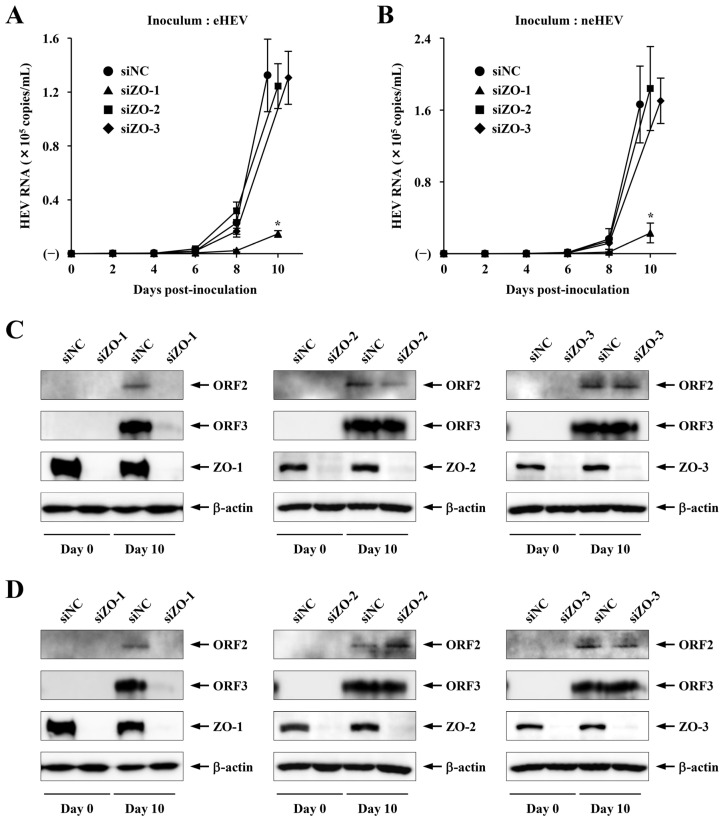
Impact of ZO-1, ZO-2, or ZO-3 silencing on HEV infection. (**A**,**B**) HEV RNA titers in the culture supernatants of eHEV-inoculated cells (**A**) and neHEV-inoculated cells (**B**) treated with siRNA targeting ZO-1, ZO-2, or ZO-3. Cells were transfected with siRNA 3 days before and 4 days after virus inoculation. Data are expressed as mean ± SD from three independent experiments. Statistically significant differences are indicated (* *p* < 0.01). (**C**,**D**) Intracellular expression of HEV ORF2 and ORF3 proteins and knockdown efficiency of siRNA specific for ZO-1, ZO-2, or ZO-3. On the specified days following inoculation with eHEV (**C**) or neHEV (**D**), cells were lysed, and the expression levels of ORF2 protein, ORF3 protein, ZO-1, ZO-2, ZO-3, and β-actin were analyzed by Western blotting with anti-ORF2 MAb, anti-ORF3 MAb, anti-ZO-1 PAb, anti-ZO-2 PAb, anti-ZO-3 MAb, and anti-β-actin MAb, respectively.

**Figure 7 pathogens-13-01130-f007:**
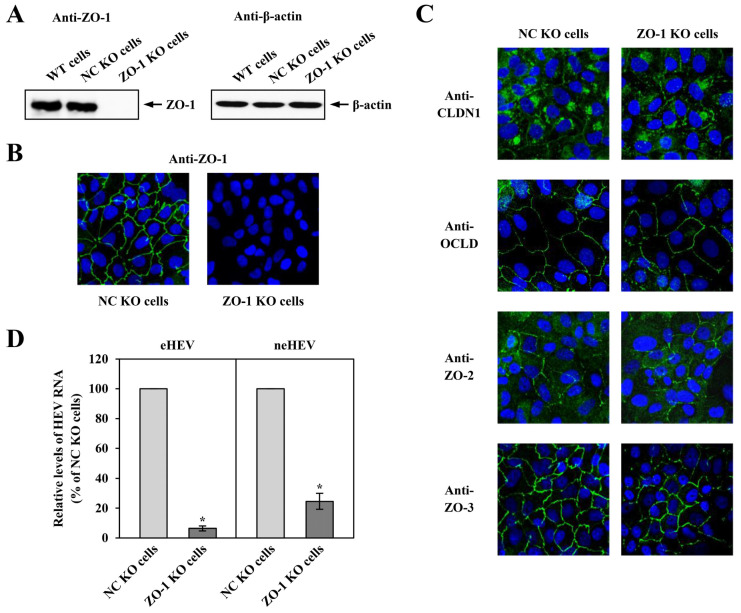
Validation of ZO-1 knockout (KO) PLC/PRF/5 cells. (**A**) Western blotting of ZO-1 and β-actin protein expression in cell lysates from wild-type (WT), NC KO, and ZO-1 KO PLC/PRF/5 cells. (**B**) Immunofluorescence staining showing ZO-1 protein expression in NC KO cells (**left panel**) and ZO-1 KO cells (**right panel**). (**C**) Immunofluorescence staining to access subcellular localization of CLDN1, OCLN, ZO-2, and ZO-3 in NC KO cells (**left panels**) and in ZO-1 KO cells (**right panels**). (**D**) Assessment of the impact of ZO-1 KO on HEV infection. NC KO and ZO-1 KO cells were inoculated with eHEV or neHEV. After 4 days, cells were collected, and intracellular HEV RNA titers in cells inoculated with eHEV (**left**) or neHEV (**right**) were quantified. Data are presented as the mean ± SD from three wells each. * *p* < 0.001.

**Figure 8 pathogens-13-01130-f008:**
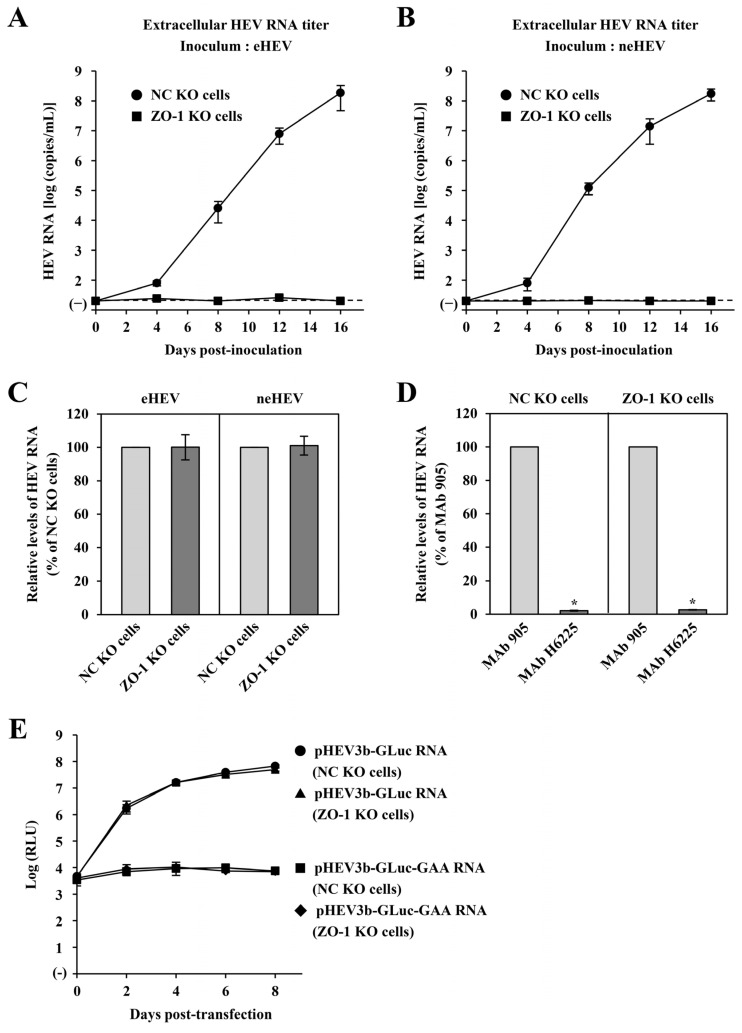
Analysis of HEV replication dynamics in ZO-1 KO cells. (**A**,**B**) Quantification of HEV titers in the culture supernatants of NC KO and ZO-1 KO cells inoculated with either eHEV (**A**) or neHEV (**B**). Cells were infected with eHEV or neHEV, and viral replication was assessed over a 16-day period. The dotted horizontal line represents the limit of detection by real-time RT-PCR used in the current study, at 2.0 × 10^1^ copies/mL. (**C**) Assessment of the adsorption efficiency of eHEV and neHEV to ZO-1 KO cells. NC KO and ZO-1 KO cells were exposed to eHEV or neHEV (2.0 × 10^4^ copies/well) at 4 °C for 1 h, followed by quantification of adsorbed viral particles. (**D**) Assessment of the adsorption efficiency of neHEV with neutralizing antibody. The neHEV (2.0 × 10^4^ copies) that had been treated with anti-ORF2 MAb (H6225; 0.2 mg/mL) or a negative control MAb (MAb 905; 0.2 mg/mL) before inoculation and then NC KO and ZO-1 KO cells were exposed to the treated neHEV at 4 °C for 1 h, followed by quantification of adsorbed viral particles. (**E**) Evaluation of HEV RNA replication efficiency in ZO-1 KO cells. NC KO and ZO-1 KO cells were transfected with either pHEV3b-GLuc RNA or the replication-deficient pHEV3b-GLuc-GAA RNA. GLuc activity was determined in culture supernatants at 2, 4, 6, and 8 dpt. Data are presented as the mean ± SD from two independent experiments. * *p* < 0.001.

**Figure 9 pathogens-13-01130-f009:**
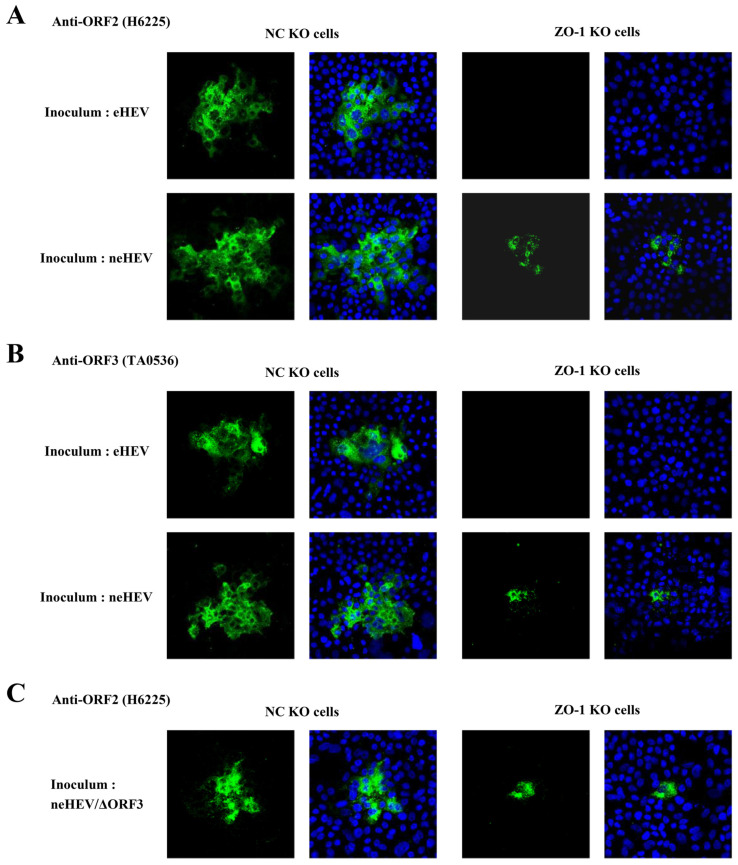
Analysis of HEV intercellular spread in ZO-1 KO cells. (**A**,**B**) Immunofluorescence staining was performed on NC KO cells (**left panels**) and ZO-1 KO cells (**right panels**) inoculated with either eHEV (**upper panels**) or neHEV (**lower panels**). Viral protein expression was detected using anti-ORF2 MAb (**A**) and anti-ORF3 MAb (**B**) at 4 dpi. (**C**) Immunofluorescence staining of NC KO cells (**left panels**) and ZO-1 KO cells (**right panels**) inoculated with neHEV/ΔORF3 to evaluate HEV ORF2 protein expression, using anti-ORF2 MAb at 4 dpi. Results shown are representative of three independent experiments.

**Figure 10 pathogens-13-01130-f010:**
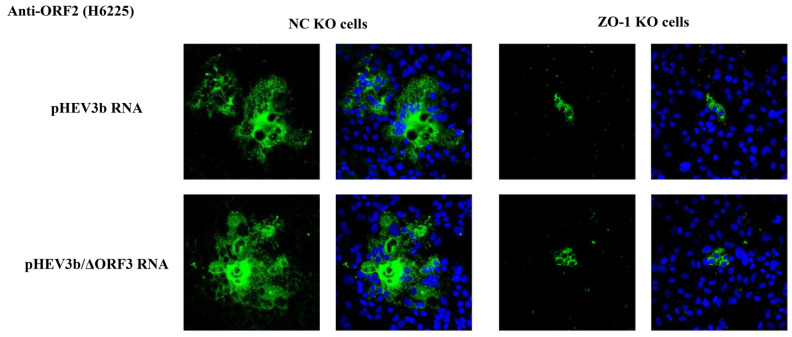
Analysis of HEV intercellular spread in ZO-1 KO cells by RNA transfection. Immunofluorescence staining was performed on NC KO cells (**left panels**) and ZO-1 KO cells (**right panels**) transfected with either pHEV3b RNA (**upper panels**) or pHEV3b/ΔORF3 RNA (**lower panels**). Viral protein expression was detected using anti-ORF2 MAb at 6 dpi. Results shown are representative of three independent experiments.

**Figure 11 pathogens-13-01130-f011:**
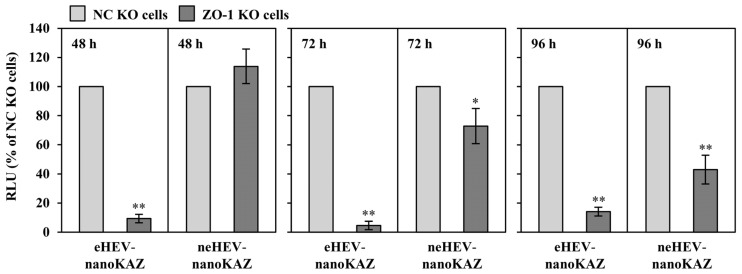
Impact of ZO-1 knockout on HEV entry. NC KO and ZO-1 KO cells were inoculated with either eHEV-nanoKAZ or neHEV-nanoKAZ, and cell lysates were collected 48, 72, and 96 h post-inoculation. Intracellular luciferase activity was quantified at the indicated times for the cells inoculated with eHEV-nanoKAZ (**left panel**) or neHEV-nanoKAZ (**right panel**). Data are presented as the mean ± SD from two independent experiments. Statistical significance was assessed using * *p* < 0.05 and ** *p* < 0.001.

## Data Availability

All data are presented in the manuscript.

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
