# Peer review of "Role of Rab13, Protein Kinase A, and Zonula Occludens-1 in Hepatitis E Virus Entry and Cell-to-Cell Spread: Comparative Analysis of Quasi-Enveloped and Non-Enveloped Forms"

_pathogens, 2024, doi:10.3390/pathogens13121130_

Round 1
Reviewer 1 Report
Comments and Suggestions for Authors
In this study, Shigeo Nagashima et al. used siRNA, Crispr KO and chemical inhibitors to elucidate the roles of host factors in HEV entry step, specifically the eHEV and neHEV entry. They demonstrated a role Rab13 and type II PKA in both type virion entry step. Further they showed that the involvement of tight junction (TJ) proteins zonula occludens-1 (ZO-1) in facilitating eHEV entry and mediating the cell-to-cell spread of neHEV in infected cells. Overall, the data are solid and provide new insights into HEV entry.
Minor:
1. In the virus adsorption assay, how many times of PBS washing were performed? The author may consider including a binding inhibitor as a positive control, such as antibody?Accordingly, the same comment for Fig 8C.
2. Line 378 and line 468: What is the meaning of “higher affinity”? Did the authors mean more effective on eHEV than on neHEV?
3. Line609-615. The authors may rephase this paragraph and explain that the purpose of using HEV-Gluc RNA transfection is to bypass virus entry step and exclusively assess viral replication.
4. In Fig 8A and 8B, ZO-1 KO reduced both eHEV and neHEV extracellular RNA titer, while in Fig 10, ZO-1 KO exclusively affect eHEV but not neHEV, please explain the difference of the phenotypes.
Author Response
Responses to Reviewer 1
Comments and Suggestions for Authors
In this study, Shigeo Nagashima et al. used siRNA, Crispr KO and chemical inhibitors to elucidate the roles of host factors in HEV entry step, specifically the eHEV and neHEV entry. They demonstrated a role Rab13 and type II PKA in both type virion entry step. Further they showed that the involvement of tight junction (TJ) proteins zonula occludens-1 (ZO-1) in facilitating eHEV entry and mediating the cell-to-cell spread of neHEV in infected cells. Overall, the data are solid and provide new insights into HEV entry.
Response: We sincerely appreciate your positive evaluation of our study and your valuable comments and suggestions. Below, we provide point-by-point responses to the minor comments.
Minor comments:
- In the virus adsorption assay, how many times of PBS washing were performed? The author may consider including a binding inhibitor as a positive control, such as antibody?Accordingly, the same comment for Fig 8C.
Response: Thank you for your insightful comments. In the virus adsorption assay, we performed five washes with PBS(-). This detail has been added to the Materials and Methods section (page 7, lines 1251-1252). To address your suggestion regarding a positive control, we conducted additional experiments using MAb H6225, a neutralizing antibody targeting the HEV capsid protein, as a positive control to inhibit virus adsorption. As a negative control, we used the unrelated MAb 905. Our results showed that MAb H6225 inhibited virus adsorption by more than 97% compared to MAb 905, demonstrating the specificity of the adsorption assay. These findings have been included in Figure 8D and are described in the Materials and Methods section (page 7, lines 1252-1254) and the Results section (page 20, lines 2199-2202).
- Line 378 and line 468: What is the meaning of “higher affinity”? Did the authors mean more effective on eHEV than on neHEV?
Response: Thank you for pointing this out. You are correct; “high affinity” in this context indicates that eHEV depends more strongly on Rab13 and PKA compared to neHEV. We have clarified this point in the revised manuscript (page 9, lines 1587-1589; page 13, lines 1748-1750).
- Line609-615. The authors may rephase this paragraph and explain that the purpose of using HEV-Gluc RNA transfection is to bypass virus entry step and exclusively assess viral replication.
Response: Thank you for your suggestion. We have revised this paragraph to clarify that the purpose of using HEV-Gluc RNA transfection was to bypass the virus entry step, allowing us to specifically assess viral replication. The updated text can be found on page 20, lines 2204-2205.
- In Fig 8A and 8B, ZO-1 KO reduced both eHEV and neHEV extracellular RNA titer, while in Fig 10, ZO-1 KO exclusively affect eHEV but not neHEV, please explain the difference of the phenotypes.
Response: Thank you for your observation. The experiments in Figure 11 (previous Figure 10) specifically assessed the efficacy of internalization using HEV-nanoKAZ at 48 h post-infection. The results showed that ZO-1 knockout selectively inhibited eHEV internalization, leading to reduced luciferase activity. In contrast, neHEV internalization was not significantly affected. However, in long-term culture with knockdown or knockout, the proliferation of both eHEV and neHEV was inhibited. For eHEV, this inhibition is attributed to impaired internalization. For neHEV, while the entry step was not affected, the cell-to-cell spread was significantly reduced, leading to an overall decrease in proliferation. These explanations have been incorporated into the Results section (page 22, lines 2306-2319; page 24, lines 2392-2401).

Reviewer 2 Report
Comments and Suggestions for Authors
The authors investigated the role of Rab13, PKA, and ZO-1 in the lifecycle of HEV infection and found that Rab13 is required for efficient HEV entry, but not RNA replication or virion release; PKA, specifically type II PKA, is involved in HEV entry but no other steps; ZO-1 may be involved in late steps during HEV entry or cell-to-cell transmission.
Although the experiments were well designed and the manuscript was well organized, I still have some concerns.
1. For the role of Rab13 for HEV, the authors did not describe in line 336 or Fig.1E whether knockdown of Rab13 significantly (p value) inhibited neHEV infection. Given that the authors measured Rab13 protein expressions at day 0 and day 10, why the authors did not measure HEV protein (ORF2, ORF3, or ORF1) at the same time? That would be another piece of evidence to support the authors’ claim. Same issue for Fig.2E, Fig.6D, Fig.6E.
2. I have big concerns for the investigation into PKA effect on HEV. Firstly, although the authors showed in Fig.3C-F that other protein kinase inhibitors did not affect HEV infection, the authors did not show that those agents at the concentrations shown in the figures had the ability to inhibit protein kinase activity. The authors have to provide the functional evidence of these agents. Same issue for Fig. 5B (type I PKA inhibitor Rp-8-Br-cAMPS), the evidence that at those concentrations this agent exerted inhibitory effect to inhibit type I PKA activity should be provided. Secondly, the authors should use knockdown of type I and type II PKA to confirm the impact on HEV infection. Chemicals may not be very specific, gene knockdown assay is still required to consolidate the conclusion.
3. Biggest concern comes from the cell-to-cell transmission results. In Fig.6A, the authors should show the knockdown efficiency for each siRNA. I don’t quite understand the results in Fig.9. Firstly, whether this assay can truly measure the cell-to-cell transmission is not conclusive yet. Some virions released out of cells can still locally infect adjacent cells, that does not account for cell-to-cell transmission, and how percentage of this local cell-free transmission contributes to the foci formation is not known. Secondly, neHEV is derived from depletion of supernatant HEV envelopes with chemicals mentioned in Methods, the only difference between neHEV and eHEV infection is the differential entry efficiency, HEV RNA replication or virus release should be identical, neHEV should perform the same with eHEV after entry into cells. That suggests the differences in the size of foci shown in Fig. 9 may only reflect the difference in late steps of virus entry between eHEV and neHEV, as suggested in Fig.10, but no difference in cell-to-cell transmission. Therefore, I will not trust any claims based on this result, like in line 762, line 772. Lastly, the ZO-1 effect on late stage of HEV entry needs direct evidence to prove.
Author Response
Responses to Reviewer 2
Comments and Suggestions for Authors
The authors investigated the role of Rab13, PKA, and ZO-1 in the lifecycle of HEV infection and found that Rab13 is required for efficient HEV entry, but not RNA replication or virion release; PKA, specifically type II PKA, is involved in HEV entry but no other steps; ZO-1 may be involved in late steps during HEV entry or cell-to-cell transmission.
Although the experiments were well designed and the manuscript was well organized, I still have some concerns.
Response: We sincerely appreciate your thoughtful comments and valuable suggestions. Below, we provide our detailed responses to each point raised.
- For the role of Rab13 for HEV, the authors did not describe in line 336 or Fig.1E whether knockdown of Rab13 significantly (p value) inhibited neHEV infection. Given that the authors measured Rab13 protein expressions at day 0 and day 10, why the authors did not measure HEV protein (ORF2, ORF3, or ORF1) at the same time? That would be another piece of evidence to support the authors’ claim. Same issue for Fig.2E, Fig.6D, Fig.6E.
Response: Thank you for your insightful comments. Following your suggestion, we have added the p-value (0.066) for Rab13 knockdown to the Results section (page 8, lines 1470-1473) and updated Figure 1C accordingly. In addition, we now include data on ORF2 and ORF3 protein expressions at days 0 and 10, which are presented in the revised Figure 1D (previous 1B), Figure 1E (previous 1C), Figure 6C (previous 6D), and Figure 6D (previous 6E). For Figure 2E, we were unable to detect the expression of ORF2 and ORF3 proteins by Western blotting because the time point (four days post-virus inoculation or RNA transfection) may have been insufficient for their detectable expression. However, as the nanoKAZ gene or HiBiT gene is inserted into the ORF1 or ORF2 coding region, respectively, the luciferase activity provides an indirect but reliable measure of viral protein production. We believe this approach supports the conclusions regarding viral replication and entry efficiency.
- I have big concerns for the investigation into PKA effect on HEV. Firstly, although the authors showed in Fig.3C-F that other protein kinase inhibitors did not affect HEV infection, the authors did not show that those agents at the concentrations shown in the figures had the ability to inhibit protein kinase activity. The authors have to provide the functional evidence of these agents. Same issue for Fig. 5B (type I PKA inhibitor Rp-8-Br-cAMPS), the evidence that at those concentrations this agent exerted inhibitory effect to inhibit type I PKA activity should be provided. Secondly, the authors should use knockdown of type I and type II PKA to confirm the impact on HEV infection. Chemicals may not be very specific, gene knockdown assay is still required to consolidate the conclusion.
Response: Thank you for your constructive feedback. To address your first point, we have included references to several published studies that validate the inhibitory effects of the drug concentrations used in our experiments. These references have been added to the Materials and Methods section (page 5, lines 760-765) and Discussion section (page 26, lines 2550-2552).
Regarding your second point, while we acknowledge the importance of gene knockdown experiments for validating the specificity of type I and type II PKA’s involvement in HEV infection, we were unable to perform siRNA knockdown for these kinases due to technical limitations. Consequently, we have decided to remove Fig. 5 from the manuscript to maintain the scientific rigor of our study.
- Biggest concern comes from the cell-to-cell transmission results. In Fig.6A, the authors should show the knockdown efficiency for each siRNA. I don’t quite understand the results in Fig.9. Firstly, whether this assay can truly measure the cell-to-cell transmission is not conclusive yet. Some virions released out of cells can still locally infect adjacent cells, that does not account for cell-to-cell transmission, and how percentage of this local cell-free transmission contributes to the foci formation is not known. Secondly, neHEV is derived from depletion of supernatant HEV envelopes with chemicals mentioned in Methods, the only difference between neHEV and eHEV infection is the differential entry efficiency, HEV RNA replication or virus release should be identical, neHEV should perform the same with eHEV after entry into cells. That suggests the differences in the size of foci shown in Fig. 9 may only reflect the difference in late steps of virus entry between eHEV and neHEV, as suggested in Fig.10, but no difference in cell-to-cell transmission. Therefore, I will not trust any claims based on this result, like in line 762, line 772. Lastly, the ZO-1 effect on late stage of HEV entry needs direct evidence to prove.
Response: Thank you for your thoughtful and detailed comments. We have carefully addressed each of the concerns raised and revised the manuscript accordingly to clarify and strengthen our findings. Below, we provide our responses to each point.
Knockdown Efficiency in Fig. 6A: According to your suggestion, we have added the results of semi-quantitative RT-PCR and a graphical representation of mRNA expression levels in Figures 5B and 5C, respectively, to demonstrate the knockdown efficiency for each siRNA.
Cell-to-Cell Transmission Analysis: We acknowledge the complexity of distinguishing between cell-free and cell-to-cell transmission. In this study, we used agarose overlays to analyze cell-to-cell spread, a method that is widely recognized as effective in suppressing the diffusion of cell-free virions while allowing direct cell-to-cell transmission. This approach has been employed in studies of various viruses (El Najjar et al., PLoS Pathog. 2016; Jin et al., PLoS Biol, 2009; Martin et al., J Virol, 2010; Timpe et al., Hepatolory, 2008). We have now emphasized this methodological aspect in the Discussion section (page 26, lines 2579-2587) to clarify its validity for assessing cell-to-cell transmission.
Impact of Virus Entry and ZO-1 on Foci Formation: To address your concern about the potential influence of late-stage virus entry on foci size differences in ZO-1 KO cells, we conducted an additional experiment using RNA transfection. This approach bypasses the virus entry step and allows us to evaluate subsequent stages of the viral life cycle. In cells transfected with wild-type RNA (pHEV3b RNA), both eHEV and neHEV were produced intracellularly, and ORF2 protein expression within infection foci was significantly reduced in ZO-1 KO cells compared to NC KO cells. Similarly, in cells transfected with an ORF3-deficient variant RNA (pHEV3b/ΔORF3 RNA), which produces only neHEV, ORF2 protein expression within foci was also significantly reduced in ZO-1 KO cells compared to NC KO cells. These findings confirm that the observed differences in foci size are attributable to virus transmission rather than the initial steps of HEV infection. We have incorporated these results into Figure 10 (new) and updated the Results section (page 22, lines 2306-2319; page 24, lines 2392-2401) to include a detailed explanation of these findings.
ZO-1’s Role in Late Stages of HEV Entry: To further elucidate the role of ZO-1 in the late stages of HEV entry, we recognize the importance of providing direct evidence. While this remains an area for future investigation, our current results indicate that ZO-1 primarily influences cell-to-cell transmission rather than late-stage entry events. We have discussed this limitation and proposed future directions in the Discussion section (page 27, lines 2783-2787).
We sincerely appreciate your valuable feedback, which has allowed us to improve the clarity and robustness of our study. We hope the revised manuscript addresses your concerns comprehensively. Please let us know if further clarification is required.

Round 2
Reviewer 2 Report
Comments and Suggestions for Authors
1. The authors need to provide raw Western blot images for ORF2 and ORF3 in Fig. 6.
2. I still think the effects of ZO-1 on the late step of the HEV entry, viral assembly and release step are not clear, those gaps should be filled in, but I do agree that ZO-1 is required for the HEV lifecycle, based on the evidence in this study.
Author Response
Comments and Suggestions for Authors
- The authors need to provide raw Western blot images for ORF2 and ORF3 in Fig. 6.
Response: Please note that original WB images for ORF2 and ORF3 in Fig. 6 “pathogens-3297541_Original images for additional experiments” have already been uploaded.
- I still think the effects of ZO-1 on the late step of the HEV entry, viral assembly and release step are not clear, those gaps should be filled in, but I do agree that ZO-1 is required for the HEV lifecycle, based on the evidence in this study.
Response: In accordance with your thoughtful comment, the sentence “However, the specific roles of ZO-1 in later stages of the HEV lifecycle, including viral assembly and release, remain unclear. Addressing these gaps will be a focus of future studies” was added in the Discussion section (Lines 2709-2711 in the manuscript with highlights).